# Molecular characteristics and laminar distribution of prefrontal neurons projecting to the mesolimbic system

Ákos Babiczky[1,2,3], Ferenc Matyas[1,2,4]*

[1]Research Centre for Natural Sciences, Budapest, Hungary; [2]Institute of Experimental Medicine, Budapest, Hungary; [3]Doctoral School of Psychology/Cognitive Science, Budapest University of Technology and Economics, Budapest, Hungary; [4]Department of Anatomy and Histology, University of Veterinary Medicine, Budapest, Hungary

**Abstract** Prefrontal cortical influence over the mesolimbic system – including the nucleus accumbens (NAc) and the ventral tegmental area (VTA) – is implicated in various cognitive processes and behavioral malfunctions. The functional versatility of this system could be explained by an underlying anatomical complexity; however, the detailed characterization of the medial prefrontal cortical (mPFC) innervation of the NAc and VTA is still lacking. Therefore, combining classical retrograde and conditional viral tracing techniques with multiple fluorescent immunohistochemistry, we sought to deliver a precise, cell- and layer-specific anatomical description of the cortico-mesolimbic pathways in mice. We demonstrated that NAc- ($mPFC_{NAc}$) and VTA-projecting mPFC ($mPFC_{VTA}$) populations show different laminar distribution (layers 2/3–5a and 5b–6, respectively) and express different molecular markers. Specifically, calbindin and Ntsr1 are specific to $mPFC_{NAc}$ neurons, while $mPFC_{VTA}$ neurons express high levels of Ctip2 and FoxP2, indicating that these populations are mostly separated at the cellular level. We directly tested this with double retrograde tracing and *Canine adenovirus type 2*-mediated viral labeling and found that there is indeed minimal overlap between the two populations. Furthermore, whole-brain analysis revealed that the projection pattern of these populations is also different throughout the brain. Taken together, we demonstrated that the NAc and the VTA are innervated by two, mostly nonoverlapping mPFC populations with different laminar distribution and molecular profile. These results can contribute to the advancement in our understanding of mesocorticolimbic functions and its disorders in future studies.

## Editor's evaluation

This study provides valuable and detailed information regarding the connectivity between the medial prefrontal cortex (mPFC) and two major projection targets, the nucleus accumbens (NAc) and the ventral tegmental area (VTA). The authors show that mPFC neurons projecting to the NAc and VTA form distinct, largely non-overlapping cell groups characterized by distribution patterns in mPFC, their layers, and gene expressions. The authors also identify useful molecular markers for these populations. Overall, this study provides a valuable and solid resource with which to investigate neural circuits involved in motivated behaviors.

## Introduction

The medial prefrontal cortex (mPFC), the nucleus accumbens (NAc), and the ventral tegmental area (VTA) are the three major elements of the mesocorticolimbic system that controls a wide range of behaviors (*Tzschentke and Schmidt, 2000*; *Russo and Nestler, 2013*; *Riga et al., 2014*). mPFC

*For correspondence:
matyasf@koki.hu

Competing interest: The authors declare that no competing interests exist.

provides the major source of glutamatergic input to the NAc (*Brog et al., 1993*; *Asher and Lodge, 2012*; *Li et al., 2018*) and to the VTA (*Geisler and Zahm, 2005*; *Mahler and Aston-Jones, 2012*; *Faget et al., 2016*). Direct mPFC innervation in the NAc has been implicated in various cognitive processes and malfunctions, such as attention regulation (*Christakou et al., 2004*), impulse control (*Feja and Koch, 2015*), addiction (*Schmidt et al., 2005*; *Peters et al., 2008*; *Seif et al., 2013*; *Domingo-Rodriguez et al., 2020*), and depression (*Vialou et al., 2014*). mPFC can also bidirectionally modulate neuronal activity in VTA, including NAc- and mPFC-projecting dopaminergic neurons (*Gariano and Groves, 1988*; *Carr and Sesack, 2000*; *Lodge, 2011*). Accordingly, the stimulation of excitatory neurons in the mPFC elicits dopamine release in the NAc via the VTA (*Taber et al., 1995*; *Karreman and Moghaddam, 1996*) and optogenetic activation of mPFC input in the VTA is reinforcing (*Beier et al., 2015*; *Pan et al., 2021*). Although, excitatory neurons in the mPFC are distributed in distinct layers and possess various projection patterns and molecular identity, it is not known how this diversity correlates to the abovementioned cortical functions.

Several well-established classification systems exist, based on anatomical, physiological, molecular, and connectivity profile of excitatory cortical neurons (*Harris et al., 2014*; *Harris et al., 2019*; *Harris and Shepherd, 2015*; *Baker et al., 2018*; *Bakken et al., 2021*; *Gao et al., 2022*). A widely accepted one divides principal neurons to three major classes according to their laminar distribution and projection pattern. Intertelencephalic (IT) cells are present in layers 2–6 (L2–6) and project to ipsi- and contralateral neocortex and striatum. Neurons of the pyramidal tract (PT, also known as extratelencephalic) class are located mostly in the L5b and innervate mostly mesencephalic and diencephalic regions. The third, corticothalamic (CT) class is composed of neurons in the L6 that innervate the thalamus. However, some studies suggest that this classification might be oversimplified and not universally applicable to all cortical areas (*Groh et al., 2010*; *Kim et al., 2015*). Indeed, a recent publication (*Gao et al., 2022*) divided prefrontal cortical neurons into even more new subtypes based on their genetic identity and connectivity. These results implicate that experiments involving cortical projection neurons embedded in the mesocorticolimbic system require a combination of cell-, layer-, and class-selective approaches to ensure appropriate precision.

The lack of wide-spread adoption of such specific experimental approaches in the mPFC might be the source of contradictions and inconsistencies present in the mesocorticolimbic literature. For instance, a number of publications (*Pinto and Sesack, 2000*; *Kim et al., 2017*; *Cruz et al., 2021*) demonstrated that NAc- and VTA-projecting neurons are mostly nonoverlapping at the cellular level. However, a recent study (*Gao et al., 2020*) found significant overlap between these populations in the anterior cingulate cortex, a major subregion of the mPFC demonstrating that all VTA-projecting neurons simultaneously project to NAc as well. Such inconsistencies could be resolved by applying integrated layer-, region-, and cell-selective approaches.

Therefore, we have begun to describe the prefrontal innervation of the NAc and VTA in a class-, layer-, region-, and cell-specific manner. We used neurochemical markers that provide an easy-to-use, consistent and biologically relevant framework to precisely delineate prefrontal cortical layers and territories. Using this framework, we report that NAc and VTA are innervated by two, rather nonoverlapping mPFC neuron populations. While NAc-innervating neurons tend to be found in the L2/3 and L5a, VTA-projecting cells are mostly localized in the L5b and L6, resembling IT and PT projection classes, respectively, which results were confirmed using layer-selective transgenic mouse lines. Accordingly, these two populations express different combination of molecular markers and have different afferent connections throughout the brain. Furthermore, we found that in comparison with primary cortical areas, the mPFC differs in several cytoarchitectural features.

## Results

### Distribution and molecular characterization of NAc-projecting mPFC cells

In order to investigate the mPFC-NAc connection, first, we injected retrograde tracers Cholera toxin B (CTB) subunit or Fluoro-Gold (FG) into the NAc (*Figure 1A–C*). Injection sites included both the core (NAcC) and shell (NAcSh) region (*Figure 1C*). Retrogradely labeled NAc-projecting mPFC cells (mPFC$_{NAc}$) were present throughout the mPFC. To identify the exact subregional distribution of mPFC$_{NAc}$ neurons, we performed multiple fluorescent immunohistochemical (IHC$_{Fluo}$) staining for

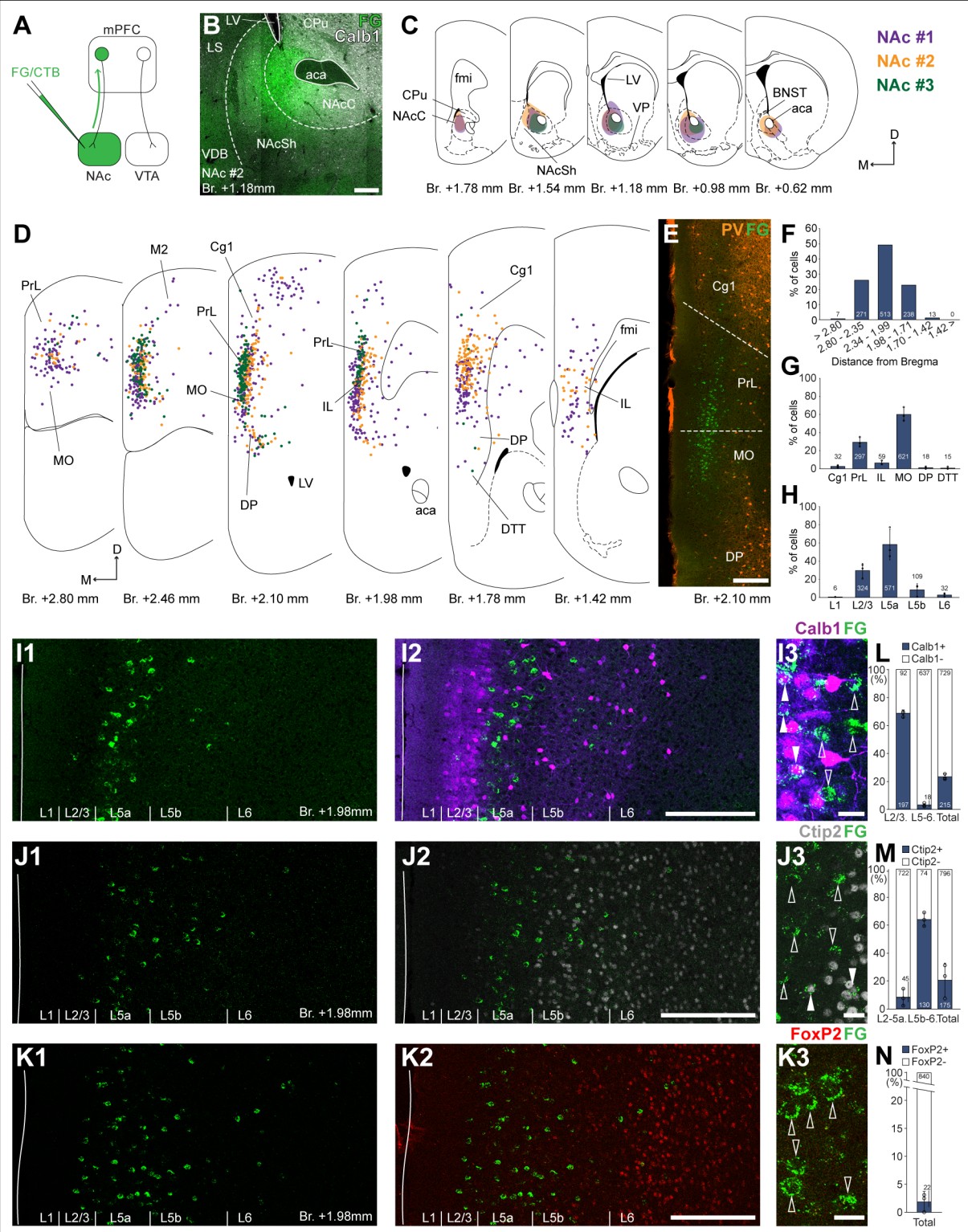

**Figure 1.** Nucleus accumbens (NAc) is innervated by L2/3 and L5 medial prefrontal cortical (mPFC) cells. (**A**) Experimental design. (**B**) A representative retrograde tracer (Fluoro-Gold [FG], green) injection site in the NAc. (**C**) Extent of injection sites in the NAc of three animals. Each case is represented with different color. (**D**) Plotted distribution of retrogradely labeled cells throughout the mPFC of the same animals as in **C** (same colors represent same animals). Each dot represents one labeled mPFC_NAc cell. (**E**) Distribution of labeled mPFC_NAc neurons in relation to parvalbumin (PV) (orange) immunofluorescent labeling outlining the PrL cortex (*Figure 1—figure supplement 1*). (**F**) Pooled anteroposterior distribution of mPFC_NAc neurons for three animals. (**G**) Distribution of mPFC_NAc cells in individual mPFC subregions. (**H**) Laminar distribution of mPFC_NAc neurons in the mPFC. (**I–K**) Confocal

*Figure 1 continued on next page*

*Figure 1 continued*

images showing the distribution of FG-labeled cells (green) in the PrL at Br. + 1.98 mm (**I1–K1**) with the counterstaining of Calb1 (purple, **I2**), Ctip2 (gray scale, **J2**), and FoxP2 (red, **K2**) (*Figure 1—figure supplement 1*). Note that most labeled cells are localized in the L2/3 (Calb1) and L5a (Ctip2). (**I3–K3**) High-magnification confocal images showing the coexpression of FG and Calb1 (**I3**), Ctip2 (**J3**), or FoxP2 (**K3**). White arrowheads indicate colabeling, empty arrowheads indicate the lack of marker expression. (**L–N**) Bar graphs showing the proportion of Calb1- (**L**), Ctip2- (**N**), and FoxP2-expressing (**M**) mPFC$_{NAc}$ cells. All data are shown as mean ± standard deviation (SD), n = 3 mice. Numbers in the bars represent cell counts and circles represent individual animal data. For detailed quantitative data see **Tables 1 and 2**. Scale bars: (**B, E, I1–K1, I2–K2**) 200 μm; (**I3–K3**) 20 μm. aca, anterior commissure, anterior part; BNST, bed nucleus of the stria terminalis; CPu, caudate putamen; fmi, forceps minor of the corpus callosum; LS, lateral septum; LV, lateral ventricle; VDB, nucleus of the vertical limb of the diagonal band; VP, ventral pallidum.

The online version of this article includes the following figure supplement(s) for figure 1:

**Figure supplement 1.** Parvalbumin (PV), Calb1, Ctip2, and FoxP2 staining define medial prefrontal cortical (mPFC) subregion borders and layers.

different molecular markers. As it was previously reported (*Mátyás et al., 2014*), parvalbumin (PV) staining delineates the dorsal and ventral borders of the prelimbic (PrL) subregion of the mPFC (*Figure 1—figure supplement 1A*, asterisk). Calbindin (Calb1) was used to define layer 2/3 (L2/3) (*van Brederode et al., 1991*; *Sun et al., 2002*) and the ventral border of the infralimbic cortex (IL), where the clearly visible L2/3 diminishes, as well as to visualize the thickening of L1, a characteristic of the deep peduncular cortex (DP) (*Akhter et al., 2014*; *Figure 1—figure supplement 1B*, number sign). COUP-TF-interacting protein 2 (Ctip2, also known as Bcl111b) was used to outline the L5b and L6 (*Arlotta et al., 2005*; *Ueta et al., 2014*; *Kim et al., 2017*; *Figure 1—figure supplement 1C*). Furthermore, forkhead box protein P2 (FoxP2) staining identifies the L6 (*Ferland et al., 2003*) and the gradual thinning and disappearance of a distinct L6 toward the ventralmost part of the mPFC (*Figure 1—figure supplement 1D*, cross).

According to the obtained molecular-based mPFC map, most mPFC$_{NAc}$ neurons were found in the medial orbital (MO; 59.99 ± 7.57%; n = 3 animals; N = 612/1042 cells), PrL (29.31 ± 5.25%; N = 297/1042 cells), and IL (6.40 ± 2.33%; N = 59/1042 cells) subregions and, to a lower extent, in the cingulate area 1 (Cg1), DP, and dorsal tenia tecta (DTT, also known as anterior hippocampal continuation) with the highest number of cells (N = 513/1042 cells) between Bregma + 2.34 and +1.99 mm (*Figure 1D–G*; *Table 1*). A relatively low number of cells were found in the primary and secondary motor (M1–M2) and the adjacent orbital cortices (*Figure 1D*). At the laminar level, the vast majority of mPFC$_{NAc}$ cells were localized in the L5a (58.64 ± 16.39%; N = 571/1042 cells) and L2/3 (29.56 ± 7.84%; N = 324/1042 cells) (*Figure 1H, I1-K1, I2-K2*; *Table 1*).

To characterize the molecular identity of mPFC$_{NAc}$ cells, we quantified their Calb1-, Ctip2-, and FoxP2-expression (*Figure 1I–N*). Our analysis revealed that about two-thirds (68.64 ± 2.62%, n = 3 animals, $N_{Calb1+/FG+}$ = 197/289 cells; *Figure 1I3, L*, *left bar*; *Table 2*) of mPFC$_{NAc}$ neurons in the L2/3 expressed Calb1, while only a small proportion did so in the L5–6 (2.87 ± 1.15%, $N_{Calb1+/FG+}$ = 18/655 cells; *Figure 1L*, *middle bar*; *Table 2*). Collectively, approximately one-fifth of all mPFC$_{NAc}$ neurons expressed Calb1 (22.78 ± 1.86%, $N_{Calb1+/FG+}$ = 215/944 cells; *Figure 2L*, *right bar*; *Table 2*). Although most of the mPFC$_{NAc}$ cells were found in the Ctip2-negative L2/3 and 5a, some cells were found in the deeper layers as well. Confocal analysis revealed that only a small proportion of superficial (i.e., L2/3–5a) cells were Ctip2 positive (8.26 ± 2.6%, n = 3 animals, $N_{Ctip2+/FG+}$ = 45/767 cells; *Figure 1J3, M*, *left bar*; *Table 2*), while in the deeper layers (i.e., L5b–6), although relatively few in number, the majority of cells expressed Ctip2 (64.1 ± 4.76%, $N_{Ctip2+/FG+}$ = 130/204 cells; *Figure 1M*, *middle bar*; *Table 2*). Collectively, approximately one-fifth of all mPFC$_{NAc}$ cells expressed Ctip2 (20.8 ± 12.1%, $N_{Ctip2+/FG+}$ = 175/971 cells; *Figure 1M*, *left bar*; *Table 2*). Finally, only a negligible number of mPFC$_{NAc}$ cells expressed FoxP2 (2.11 ± 1.84%, n = 3 animals, $N_{FoxP2+/FG+}$ = 22/862 cells, *Figure 1K3, N*; *Table 2*).

Altogether, retrograde tracing experiments revealed that mPFC$_{NAc}$ neurons were mostly localized in the L2/3 and 5a of the PrL, MO, and IL cortices. Approximately one-fifth of these cells express Calb1 – most of them are localized in the L2/3, where Calb1 expression is higher (~70%), and another one-fifth express Ctip2, mostly in the L5b–6.

## Distribution and molecular characterization of VTA-projecting mPFC cells

Next, we investigated the distribution of VTA-projecting neurons in the mPFC (mPFC$_{VTA}$). We used the previously described retrograde tracing approach in the VTA (*Figure 2A*) identified with IHC$_{Fluo}$

**Table 1.** Anteroposterior, subregional, and laminar distribution of mPFC$_{NAc}$ and mPFC$_{VTA}$ neurons ($n$ = 3–3 mice).

| | | mPFC$_{NAc}$ | mPFC$_{VTA}$ |
|---|---|---|---|
| | >2.80 mm | 7<br>0.67% | 23<br>1.22% |
| | 2.80–2.35 mm | 271<br>26.01% | 221<br>11.77% |
| | 2.34–1.99 mm | 513<br>49.23% | 530<br>28.22% |
| | 1.98–1.71 mm | 238<br>22.84% | 864<br>46.01% |
| | 1.70–1.42 mm | 13<br>1.25% | 115<br>6.12% |
| Anteroposterior distribution (Bregma level) | <1.42 mm | 0<br>0.00% | 125<br>6.66% |
| | Cg2 | 0, 0, 0<br>0 ± 0% | 26, 16, 22<br>3.88 ± 2.33% |
| | Cg1 | 2, 26, 4<br>2.46 ± 1.38% | 53, 42, 73<br>9.57 ± 3.71% |
| | PrL | 49, 194, 54<br>29.31 ± 5.26% | 95, 257, 153<br>26.68 ± 8.63% |
| | IL | 10, 35, 14<br>6.40 ± 2.33% | 20, 67, 88<br>8.63 ± 3.25% |
| | MO | 135, 404, 82<br>59.99 ± 7.57% | 71, 128, 199<br>20.57 ± 4.46% |
| | DP | 2, 16, 0<br>1.11 ± 1.16% | 68, 111, 130<br>16.54 ± 0.75% |
| Subregional distribution | DTT | 0, 15, 0<br>0.72 ± 1.26% | 64, 86, 109<br>14.12 ± 1.98% |
| | L1 | 1, 4, 1<br>0.58 ± 0.07% | 1, 0, 0<br>0.08 ± 0.15% |
| | L2/3 | 71, 221, 32<br>29.55 ± 7.84% | 30, 9, 9<br>3.33±3.66 |
| | L5a | 92, 360, 119<br>58.64 ± 16.39% | 10, 7, 5<br>1.38 ± 1.00% |
| | L5b | 25, 83, 1<br>8.43 ± 6.75% | 194, 434, 466<br>56.82 ± 6.91% |
| Laminar distribution | L6 | 9, 22, 1<br>2.79 ± 1.98% | 162, 257, 294<br>38.38 ± 2.25% |
| Total cell count | | 198, 690, 154 | 397, 707, 774 |

Note that anteroposterior data have been pooled in both groups.

against tyrosine hydroxylase (TH; *Figure 2B, C*; *Oades and Halliday, 1987*; *Morales and Margolis, 2017*). Most mPFC$_{VTA}$ neurons were localized in the PrL (26.68 ± 8.63%; $n$ = 3 animals; $N$ = 505/1878 cells), MO (20.57 ± 4.46%; $N$ = 398/1878 cells), DP (16.54 ± 0.75%; $N$ = 309/1878 cells), DTT (14.12 ± 1.98%; $N$ = 259/1878 cells), as well as in the Cg1 (9.57 ± 3.71%; $N$ = 168/1878 cells), IL (8.63 ± 3.25%; $N$ = 175/1878 cells), and Cg2 (3.88 ± 2.33%; $N$ = 64/1878 cells) cortices, with the highest density ($N$ = 864/1878 cells) between Bregma +1.98 and +1.71 mm (*Figure 2D–G*; *Table 1*). There were also several labeled cells in the adjacent orbital and motor cortices (*Figure 2D*).

Regarding their laminar distribution, we found that most of mPFC$_{VTA}$ cells formed two main clusters (*Figure 2* D, I1–K1, I2–K2): one in the L5b (56.82 ± 6.91%; $N$ = 1094/1878 cells) most prominently in the PrL, MO, and Cg1–2 cortices and another in the L6 (38.38 ± 2.25%; $N$ = 713/1878 cells) of the IL,

**Table 2.** Proportion of FoxP2-, Ctip2-, and Calb1-expressing neurons in the mPFC$_{NAc}$ population (n = 3 mice).

**mPFC$_{NAc}$**

| | Calb1 | | |
|---|---|---|---|
| Layers | # FG+ /animal | # Calb1+ /animal | % Calb1+ (AVG ± SD) |
| L2/3 | 128, 101, 60 | 84, 71, 42 | 68.6 ± 2.6% |
| L5a-5b-6 | 287, 200, 168 | 7, 4, 7 | 2.9 ± 1.1% |
| Total | 415, 301, 228 | 91, 75, 49 | 22.8 ± 1.9% |
| | Ctip2 | | |
| Layers | # FG+ /animal | # Ctip2+ /animal | % Ctip2+ (AVG ± SD) |
| L2/3-5a | 62, 356, 349 | 9, 9, 27 | 8.26 ± 6.01% |
| L5b-6 | 37, 29, 138 | 22, 20, 88 | 64.1 ± 4.76% |
| Total | 99, 385, 487 | 31, 29, 115 | 20.8 ± 12.1% |
| | FoxP2 | | |
| Layers | # FG+ /animal | # FoxP2+ /animal | % FoxP2+ (AVG ± SD) |
| L2/3-5a-5b | 288, 357, 173 | 5, 10, 0 | 1.5 ± 1.4% |
| L6 | 20, 24, 0 | 4, 3, 0 | -* |
| Total | 308, 381, 173 | 9, 13, 0 | 2.1 ± 1.8% |

#, number of labeled cells.

Calb1, Ctip2, and FoxP2-immunostainings were used to define the cortical layers in mPFC.

*When total cell count was <10, percentage was not calculated.

DP, and DTT (*Figure 2H*; *Table 1*). The separation of these two mPFC$_{VTA}$ clusters was most prominent between Bregma +1.2 and +1.8 mm, as it was also shown in previous publications (*Geisler and Zahm, 2005*; *Mahler and Aston-Jones, 2012*).

Higher magnification confocal analysis revealed that only a marginal proportion (1.31 ± 0.5%, n = 3 animals, $N_{Calb1+/FG+}$ = 15/1165 cells, *Figure 2I3, L*; *Table 3*) of all mPFC$_{VTA}$ cells expressed Calb1. We also quantified the Ctip2expression of mPFC$_{VTA}$ neurons and found that the vast majority of these cells express Ctip2 (95.07 ± 0.6%, n = 3 animals, $N_{Ctip2+/FG+}$ = 481/506 cells; *Figure 2J3, M*; *Table 3*). This finding is in accordance with previous results (*Kim et al., 2017*) showing *CTIP2* gene enrichment in mPFC$_{VTA}$ neurons. Finally, most L6 mPFC$_{VTA}$ cells expressed FoxP2 (78.86 ± 8.79%, n = 3 animals, $N_{FoxP2+/FG+}$ = 761/951 cells; *Figure 2K3, N, middle bar*; *Table 3*). On the other hand, in the superficial layers (L2/3–L5), only a small proportion (8.69 ± 2.13%, $N_{FoxP2+/FG+}$ = 77/920 cells; *Table 3*) of mPFC$_{VTA}$ cells were FoxP2-positive (*Figure 2N, left bar*). In total, about half of all mPFC$_{VTA}$ neurons expressed FoxP2 (45.93 ± 17.15%, $N_{FoxP2+/FG+}$ = 838/1871 cells; *Figure 2N, right bar*; *Table 3*).

Taken together, using retrograde tracing experiments we identified two major clusters of mPFC$_{VTA}$ neurons distributed throughout the mPFC: one FoxP2-, and most probably Ctip2-expressing population localized mostly in the L6 (approximately half of all neurons); and one, mostly FoxP2-negative, but Ctip2-positive population in the layer 5b.

## Utility of Cre mouse lines to label mPFC neurons in a layer-selective manner

We found retrogradely labeled mPFC$_{NAc}$ and mPFC$_{VTA}$ neurons in all cellular layers of the mPFC in varying densities. Next, we sought to confirm the laminar organizations of the projecting cells using transgenic mice expressing Cre-recombinase enzyme in a layer-selective manner. We used the following layer-specific Cre-expressing mouse strains: *Calb1-* (L2/3), *Retinol Binding Protein 4-* (*Rbp4*; L5), *Neurotensin Receptor 1* (*Ntsr1*; L6), and *FoxP2-Cre* (L6) (*van Brederode et al., 1991*; *Hof et al., 1999*; *Sun et al., 2002*; *Ferland et al., 2003*; *Molyneaux et al., 2007*; *Harris et al., 2014*; *Harris et al., 2019*; *Sundberg et al., 2018*; *Callaway, 2021*; *Matho et al., 2021*; *Muñoz-Castañeda et al.,*

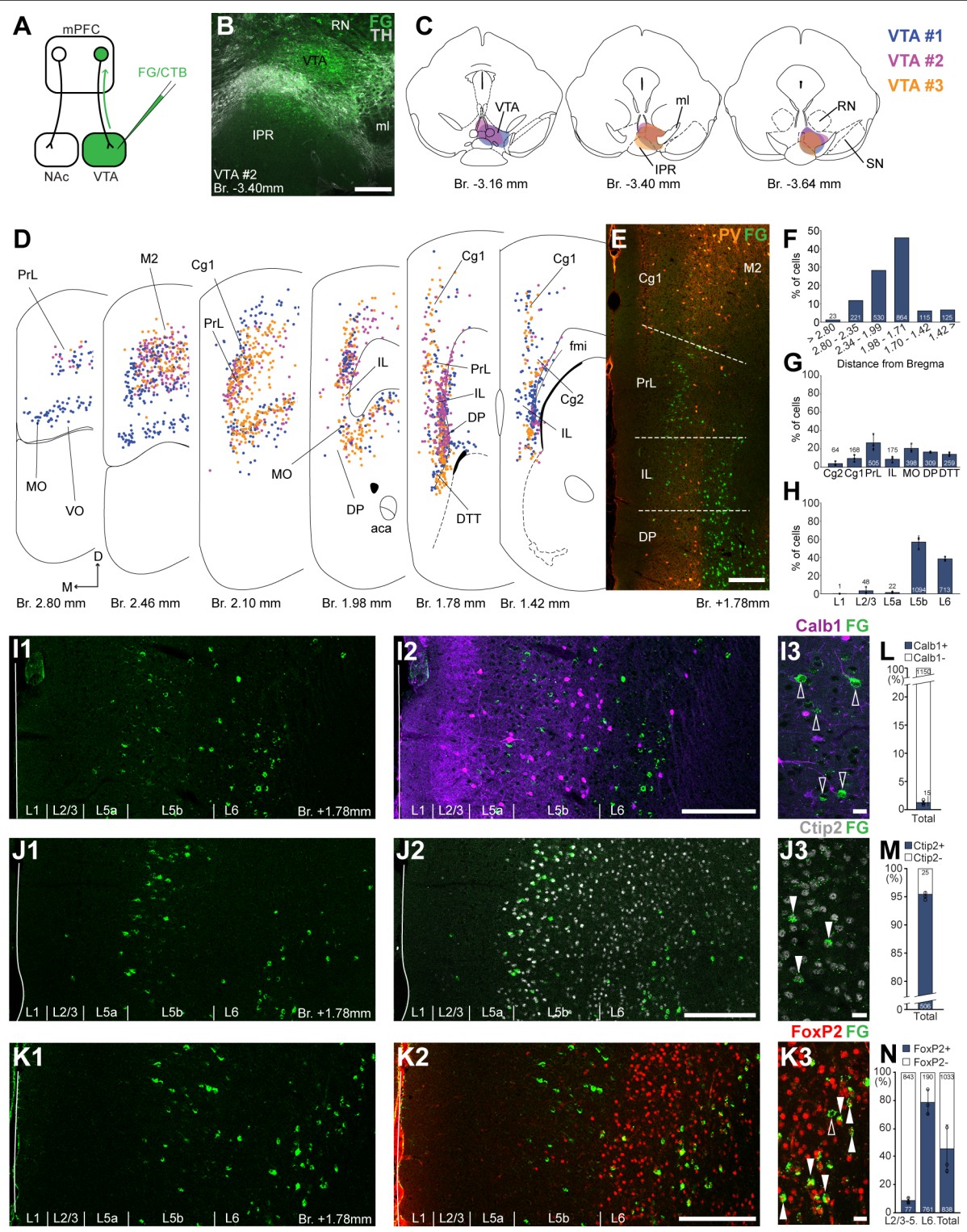

**Figure 2.** Ventral tegmental area (VTA) is innervated by two medial prefrontal cortical (mPFC) cell clusters. (**A**) Experimental design. (**B**) A representative retrograde tracer (Fluoro-Gold [FG], green) injection site in the VTA. (**C**) Full extent of the injection sites in the VTA in three animals. Each case is represented with different color. (**D**) Plotted distribution of retrogradely labeled neurons throughout the mPFC of the same animals as in (**C**) (same colors represent same animals). Each dot represents one labeled mPFCvTA cell. (**E**) Distribution of labeled neurons in the mPFC in relation to parvalbumin (PV) (orange) immunofluorescent labeling outlining the PrL cortex. (**F**) Pooled anteroposterior distribution of mPFCvTA neurons for three animals. (**G**) Distribution of mPFCvTA cells in individual mPFC subregions. (**H**) Laminar distribution of mPFCvTA neurons in the mPFC. (**I–K**) Confocal images showing

*Figure 2 continued on next page*

*Figure 2 continued*

the layer-specific distribution of FG-labeled cells (green) in the PrL (**I1–K1**) with counterstaining of Calb1 (purple, **I2**), Ctip2 (gray scale, **J2**), and FoxP2 (red, **K2**). Note that the labeled cells are almost exclusively localized in the L5b (Ctip2) and L6 (Ctip2 + FoxP2) layers. (**I3–K3**) High-magnification confocal images showing the coexpression of FG and Calb1 (**I3**), Ctip2 (**J3**), or FoxP2 (**K3**). White arrowheads indicate colabeling, empty arrowheads indicate the lack of marker expression. Bar graphs showing the proportion of Calb1- (**L**), Ctip2- (**M**), and FoxP2-expressing (**N**) mPFC$_{VTA}$ cells. All data are shown as mean ± standard deviation (SD), $n$ = 3 mice. Numbers in the bars represent cell counts and circles represent individual animal data. For detailed quantitative data see *Tables 1 and 3*. Scale bars: (**B**, **E**, **I1–K1**, **I2–K2**) 200 μm; (**I3–K3**) 20 μm. aca, anterior commissure, anterior part; fmi, forceps minor of the corpus callosum; IPR, interpeduncular nucleus, rostral subnucleus; ml, medial lemniscus; RN, red nucleus; SN, substantia nigra; VO, ventral orbital cortex.

*2021*) in combination with Cre-dependent adeno-associated viral vectors (AAVs) (*Figure 3A–D*). Furthermore, we used a *Thymocyte differentiation antigen 1* (*Thy1*)-*Cre* mouse line as control, in which Cre enzyme is expressed in all pyramidal neurons, regardless of their laminar localization (*Figure 3E*).

Virally labeled cell bodies in all strains were primarily found in the PrL, IL, Cg1–2, MO, and, to a lower extent, in the DP, the ventromedial M2, the dorsal part of the DTT and the medial part of the VO cortex (*Figure 3A–E*) in good correspondence with the distribution of the retrogradely labeled mPFC$_{NAc}$ and mPFC$_{VTA}$ neurons (*Figures 1 and 2*). Note that viral expression was always analyzed after IHC enhancement of eYFP/mCherry, because this method revealed structures – mostly thin axon branches, but also some cell bodies – and fine details (e.g., dendritic spines) otherwise not detectable (see Methods) (*Figure 3—figure supplement 1*).

Since the majority of previous publications describing cortical layer-specific markers focused on primary cortical areas, we compared the expression pattern of virally labeled cells in the mPFC – a higher-order cortical region (*Figure 3—figure supplement 2A–E*) – and in the primary motor cortex (M1, *Figure 3—figure supplement 2F-J*) – a primary frontal cortical area – in each mouse strain. Labeled cells in the *Calb1-Cre* animals showed similar distribution in both cortical areas: most of them were found in the L2/3 (*Muñoz-Castañeda et al., 2021*) with scattered cells – most likely cortical interneurons (*Staiger et al., 2004*) – in other cortical layers (*Figure 3—figure supplement 2A, F*).

**Table 3.** Proportion of FoxP2-, Ctip2-, and Calb1-expressing neurons in the mPFC$_{VTA}$ population ($n$ = 3 mice).

**mPFC$_{VTA}$**

| | Calb1 | | |
|---|---|---|---|
| Layers | # FG+ /animal | # Calb1+ /animal | % Calb1+ (AVG ± SD) |
| L2/3 | 4, 2, 5 | 4, 2, 2 | -* |
| L5a-5b-6 | 371, 452, 331 | 3, 2, 2 | -* |
| Total | 375, 454, 336 | 7, 4, 4 | 1.3 ± 0.5% |
| | Ctip2 | | |
| Layers | # FG+ /animal | # Ctip2+ /animal | % Ctip2+ (AVG ± SD) |
| L2/3-5a | 1, 2, 1 | 1, 0, 0 | -* |
| L5b-6 | 152, 163, 187 | 144, 158, 178 | 95.6 ± 1.2% |
| Total | 153, 165, 188 | 145, 158, 178 | 95.1 ± 0.6% |
| | FoxP2 | | |
| Layers | # FG+ /animal | # FoxP2+ /animal | % FoxP2+ (AVG ± SD) |
| L2/3-5a-5b | 162, 347, 411 | 17, 22, 38 | 8.7 ± 2.1% |
| L6 | 393, 283, 275 | 347, 219, 195 | 78.9 ± 8.8% |
| Total | 555, 630, 686 | 364, 241, 233 | 45.9 ± 17.2% |

#, number of labeled cells.
Calb1, Ctip2, and FoxP2-immunostainings were used to define the cortical layers in mPFC.
*When total cell count was <10, percentage was not calculated.

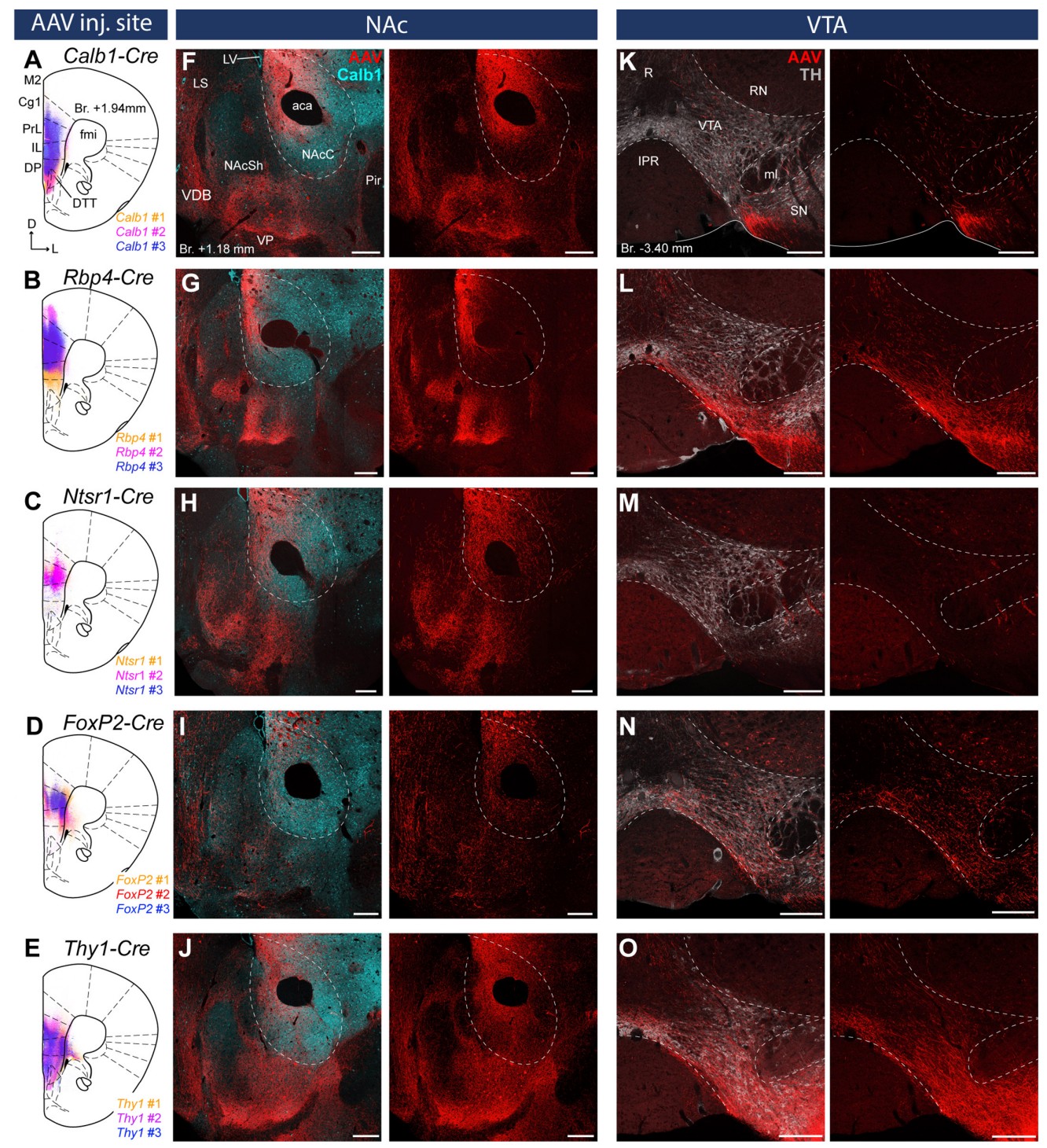

**Figure 3.** Distinct nucleus accumbens (NAc) and ventral tegmental area (VTA) innervation by genetically identified medial prefrontal cortical (mPFC) cell populations. Delineation of AAV-DIO-mCherry/eYFP injection sites in the mPFC of the *Calb1-* (**A**), *Rbp4-* (**B**) *Ntsr1-* (**C**) *FoxP2-* (**D**), and *Thy1-Cre* (**E**) strains (*n* = 3 mice in each strain). Viral labeling was always analyzed after immunohistochemical enhancement (*Figure 3—figure supplement 1*). For higher magnification distribution of labeled cells in the mPFC and M1 see *Figure 3—figure supplement 2*. Confocal images showing virally labeled prefrontal axons (red) in the NAc of *Calb1-* (**F**), *Rbp4-* (**G**), *Ntsr1-* (**H**), *FoxP2-* (**I**) and *Thy1-Cre* (**J**) mouse strains. Calb1 (cyan) immunofluorescent staining was used to identify the NAc. (**K–O**) Distribution of labeled axons (red) from the same animals, respectively, in the VTA defined with TH staining (gray scale). Scale bars: 200 μm. aca, anterior commissure, anterior part; fmi, forceps minor of the corpus callosum; IPR, interpeduncular nucleus, rostral

*Figure 3 continued on next page*

*Figure 3 continued*

subnucleus; LS, lateral septum; LV, lateral ventricle; ml, medial lemniscus; Pir, piriform cortex; R, raphe; RN, red nucleus; SN, substantia nigra; VDB, nucleus of the vertical limb of the diagonal band; VP, ventral pallidum.

The online version of this article includes the following figure supplement(s) for figure 3:

**Figure supplement 1.** IHC enhancement is necessary for reliable detection of viral fluorescent signal.

**Figure supplement 2.** Layer-specific *Cre* mouse lines reveal different laminal distribution of medial prefrontal cortical (mPFC) and primary motor cortical cells.

Interestingly, Rbp4- and Ntsr1-expressing cells showed somewhat different distribution in the two cortical regions (*Figure 3—figure supplement 2B, C, G, H*). In the *Rbp4-Cre* strain, virally labeled cells in the mPFC were found to some extent in the L2/3 – especially in the ventral part of the mPFC, in the IL and DP – besides the well-known L5 location. In the M1, only the L5 population was present (*Callaway, 2021*; *Muñoz-Castañeda et al., 2021*; *Figure 3—figure supplement 2B, G*). In the *Ntsr1-Cre* animals, no virally labeled neurons were found in the L6 in the mPFC, only in the L5a (*Figure 3—figure supplement 2C*). In the M1 cortex, Ntsr1-expressing labeled cells were found exclusively in the L6, as it was previously reported (*DeNardo et al., 2015*; *Tasic et al., 2016*; *Sundberg et al., 2018*; *Callaway, 2021*; *Muñoz-Castañeda et al., 2021*; *Figure 3—figure supplement 2H*). Regarding the *FoxP2-Cre* strain, we found that labeled cells were most abundant in the L6 in both cortical regions examined (*Figure 3—figure supplement 2D, I*), however, in the mPFC we found visually more virally transduced neurons in the L5 compared to M1. In the *Thy1-Cre* animals we did not observe any difference between the two cortical regions: AAV transduced cells were found in all cellular layers of the mPFC (*Figure 3—figure supplement 2E*) and the M1 (*Figure 3—figure supplement 2J*).

Together, these results show that these mouse strains can be used to label and investigate distinct layers of prefrontal cell populations, confirming previous findings (*van Brederode et al., 1991*; *Hof et al., 1999*; *Sun et al., 2002*; *Ferland et al., 2003*; *Molyneaux et al., 2007*; *Harris et al., 2014*; *Harris et al., 2019*; *Sundberg et al., 2018*; *Callaway, 2021*; *Matho et al., 2021*; *Muñoz-Castañeda et al., 2021*). However, in some cases (*Rbp4-*, *Ntsr1-*, and *FoxP2-Cre*) the distribution of labeled neurons was somewhat different in the mPFC compared to M1.

## Layer-selective prefrontal cortical innervation of the NAc and VTA

After validating the use of these Cre mouse strains and AAV vectors to label mPFC neuron populations in a layer-selective manner, we sought to explore their projection patterns in the NAc and VTA. In order to do this, we performed confocal microscopy combined with multiple $IHC_{Fluo}$ in tissue samples taken from the mPFC animals described in the previous section.

In the *Calb1-Cre* strain – where viral transduced cells were confined to the L2/3 – labeled axons were found in the NAc (*Figure 3F*) but not in the VTA (*Figure 3K*). These results are in accordance with our retrograde tracing results showing that a high proportion of $mPFC_{NAC}$ neurons in the L2/3 express Calb1 (*Figure 1*), and the lack of $mPFC_{VTA}$ cells in the Calb1-rich layer 2/3 (*Figure 2*). In the *Rbp4-Cre* animals (L2/3–5), AAV-labeled axons were found both in the NAc (*Figure 3G*) and VTA (*Figure 3L*) also confirming our retrograde tracing results (*Figures 1 and 2*). *Ntsr1-Cre*-expressing cells – localized in the L5a (*Figure 3—figure supplement 1C*), where most of the $mPFC_{NAc}$ neurons were found previously (*Figure 1*) – projected to the NAc with visually dense arborization (*Figure 3H*) but avoided the VTA (*Figure 3M*). In the *FoxP2-Cre* strain (L6), only a small number of AAV-labeled axons was present in the NAc (*Figure 3I*), while a relatively dense arborization of labeled axons was found in the VTA (*Figure 3N*). This is in good accordance with our previous findings demonstrating that only a marginal proportion of $mPFC_{NAC}$ neurons express FoxP2 (*Figure 1*), while almost half of all $mPFC_{VTA}$ cells does so (*Figure 2*). Finally, in the control *Thy1-Cre* strain we observed dense axonal arborization both in the NAc (*Figure 3J*) and VTA (*Figure 3O*).

Taken together, our classical retrograde and cell type-specific anterograde viral tracing experiments revealed that $mPFC_{NAc}$ and $mPFC_{VTA}$ neuron populations are mostly separated in the L2/3–5a and L5b–6, respectively, although this separation is not exclusive. Conversely, these populations seem to overlap in the L5, but it is not clear whether a single mPFC neuron projects to both targets simultaneously or shows target selectivity.

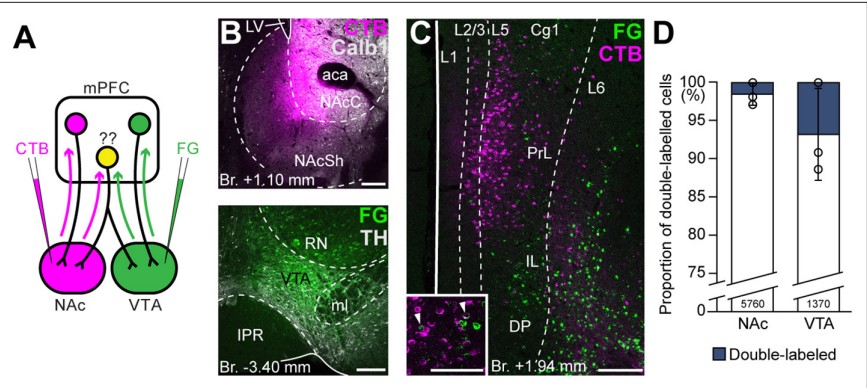

**Figure 4.** Ventral tegmental area (VTA) and nucleus accumbens (NAc) are mostly innervated by nonoverlapping medial prefrontal cortical (mPFC) cell populations. (**A**) Experimental design of double retrograde tracing experiments. (**B**) Representative CTB (magenta) injection site in the NAc (top) and Fluoro-Gold (FG) (green) in the VTA (bottom). (**C**) High-magnification confocal image showing the distribution of mPFC_NAc (magenta) and mPFC_VTA cells (green) in the mPFC. Inset shows higher magnification of the same slice with arrowheads representing double-labeled cells. (**D**) Only a small proportion of labeled mPFC cells innervated both VTA and NAc. All data are shown as mean ± standard deviation (SD), *n* = 3 mice. Exact cell counts are written in the bars. For detailed quantitative data see *Table 4*. Scale bars: 200 μm, (**C**) inset: 100 μm. aca, anterior commissure, anterior part; IPR, interpeduncular nucleus, rostral subnucleus; LV, lateral ventricle; ml, medial lemniscus; RN, red nucleus.

## NAc- and VTA-projecting mPFC populations are mostly nonoverlapping

Next, to answer the open question whether a single mPFC neuron can innervate the NAc and VTA simultaneously or not, we carried out two independent experiments to clarify this issue (*Figures 4 and 5*).

First, we performed double retrograde tracings with FG and CTB (interchangeably) from the NAc and the VTA (*Figure 4A, B*) and investigated the overlap of the labeled populations in the mPFC (*Figure 4C*). Our results showed that only a small proportion of all cells contained both tracers (NAc + VTA/VTA = 6.78 ± 5.97%, NAc + VTA/NAc = 1.54 ± 1.40%, NAc + VTA/total = 1.26 ± 1.12%; $N_{NAc}$ = 269, 2940, 2551 cells, $N_{VTA}$ = 111, 489, 770 cells, $N_{VTA+NAc}$ = 0, 55, 70 cells; *n* = 3 mice; *Figure 4D*; *Table 4*) and most of them were found in the L5.

Although these results indicate that mPFC_NAc and mPFC_VTA populations are mostly nonoverlapping at the cellular level, we considered that double retrograde technique tends to underestimate the actual proportion of multiple projecting cells. Therefore, we also applied an intersectional viral tracing approach to clarify the target selectivity of mPFC neurons. We injected *Canine adenovirus type 2* carrying *Cre-recombinase* gene (*CAV2-Cre*) into the NAc or VTA, and Cre-dependent AAV-DIO-mCherry into the mPFC (*Figure 5A–C*), a technique that was previously shown to be suitable to label cortico-tegmental and cortico-accumbal pathways (*Beier et al., 2015*; *Kerstetter et al., 2016*; *Kim et al., 2017*; *Cruz et al., 2021*). Using this method, we could selectively label mPFC_NAc and mPFC_VTA neurons with their entire axonal arborization, including collaterals projecting to other brain regions. After confirming that the injection sites were correctly positioned in the NAc or VTA (see Methods, *Figure 5B1, C1*) and in the mPFC (*Figure 5B2, C2*), we compared the projection pattern of mPFC_NAc and mPFC_VTA neurons both in the NAc and the VTA (*Figure 5D, E*). We found that mPFC_NAc axons were abundant in the NAc (*Figure 5D*, left), while only a few labeled axons were present in the VTA (*Figure 5D*, right). Conversely, mPFC_VTA neurons sent only sparse innervation to the NAc (*Figure 5E*, left), but we found dense innervation in the VTA (*Figure 5E*, right).

To quantify these results, we applied high-magnification confocal imaging (×63) to measure and compare the relative axon densities (RADs) in the two target areas. This quantitative analysis showed that mPFC_NAc neurons innervated the NAc almost tenfold stronger than the VTA (RAD_(VTA/NAc) = 0.11 ± 0.06; *n* = 3 animals; *Figure 5H*; *Table 5*; *Figure 5—source data 1*). On the other hand, mPFC_VTA cells innervated preferentially the VTA as opposed to the NAc (RAD_(VTA/NAc) = 3.45 ± 0.41; *n* = 3 animals; *Figure 5H*; *Table 5*; *Figure 5—source data 1*).

As controls, we used *Rbp4*- (mPFC_Rbp4) and *Thy1-Cre* (mPFC_Thy1) animals from the previous viral tracing experiments (*Figure 3*), since these two cell populations innervated both the NAc and VTA

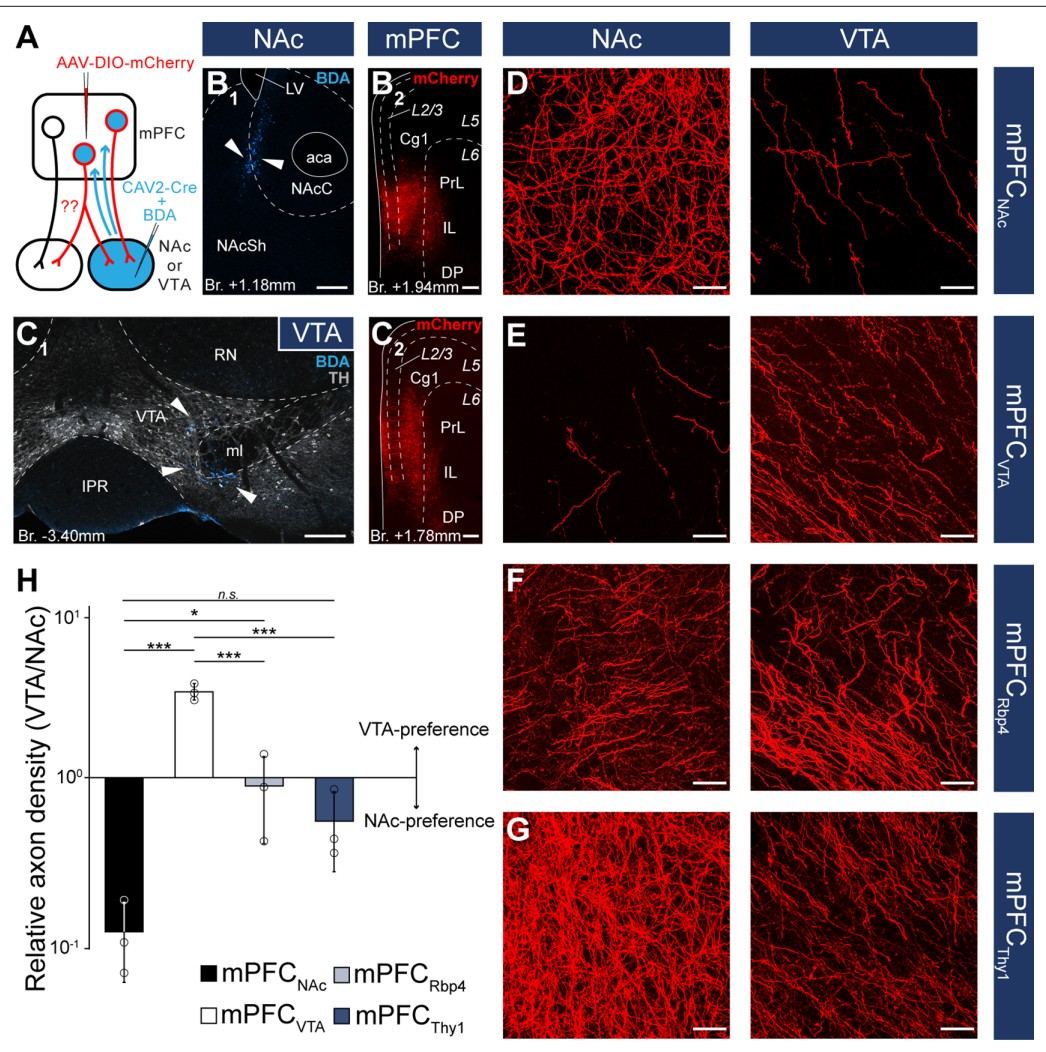

**Figure 5.** Nucleus accumbens (NAc) or ventral tegmental area (VTA) preference of medial prefrontal cortical (mPFC) cells. (**A**) Experimental design of *CAV2-Cre*-mediated viral tracing experiments. BDA was used to visualize the exact location of injection sites. (**B**) Representative *CAV2-Cre* + 5% BDA (cyan) injection site (**B1**) in the NAc and AAV-DIO-mCherry (red) injection site (**B2**) in the mPFC of the same animal. (**C**) Representative *CAV2-Cre* + BDA (cyan) injection site (**C1**) in the VTA counterstained with TH (grayscale) and AAV-DIO-mCherry (red) injection site (**C2**) in the mPFC of the same animal. (**D–G**) High-magnification confocal images showing the distribution of mCherry (red)-labeled axons in the NAc (left) and the VTA (right) in a mPFC_{NAc} (**D**), mPFC_{VTA} (**E**), mPFC_{Rbp4} (**F**); same animal as in *Figure 3C, H, M* and *Figure 3—figure supplement 1B* and mPFC_{Thy1} (**G**); same animal as in *Figure 3F, K, P* and *Figure 3—figure supplement 1E* animal. (**H**) Quantification of relative axon density (RAD) in the mPFC_{NAc}, mPFC_{VTA}, mPFC_{Rbp4}, and mPFC_{Thy1} animals $F_{(3, 8)}$ = 55.56; p = 0.000011; mPFC_{NAc} vs. mPFC_{VTA}, p = 0.0000026; mPFC_{NAc} vs. mPFC_{Rbp4}, p = 0.028; mPFC_{NAc} vs. mPFC_{Thy1}, p = 0.18; mPFC_{VTA} vs. mPFC_{Rbp4}, p = 0.000018; mPFC_{VTA} vs. mPFC_{Thy1}, p = 0.0000072; one-way analysis of variance (ANOVA), least significant difference (LSD) post hoc test; *p < 0.05; ***p < 0.001; n.s., not significant. All data are shown as mean ± standard deviation (SD), n = 3 mice in each group. For detailed quantitative data see *Table 5* and *Figure 5—source data 1*. Scale bars: (**B–C**) 200 µm, (**D–G**) 20 µm. aca, anterior commissure, anterior part; BDA, biotinylated dextran amine; IPR, interpeduncular nucleus, rostral subnucleus; LV, lateral ventricle; ml, medial lemniscus; RN, red nucleus.

The online version of this article includes the following source data for figure 5:

**Source data 1.** Detailed quantitative data for axon density analysis.

**Table 4.** Quantification of double retrograde tracing experiments ($n = 3$ mice).

| | NAc* | VTA* | Total* | Double labeled |
|---|---|---|---|---|
| # /animal | 269, 2995, 2621 | 111, 544, 840 | 380, 3374, 3251 | 0, 55, 70 |
| % double labeled | 1.5 ± 1.4% | 6.8 ± 6.0% | 1.3 ± 1.1% | – |

#, number of labeled cells.
*Including double-labeled cells.

intensively (*Figure 5F, G*). Our analysis revealed that mPFC$_{Rbp4}$ cells innervated both regions similarly (RAD$_{(VTA/NAc)}$ = 0.88 ± 0.49; $n = 3$ animals; *Figure 5F, H*; *Table 5*; *Figure 5—source data 1*), while mPFC$_{Thy1}$ cells tended to innervate NAc slightly more intensively (RAD$_{(VTA/NAc)}$ = 0.53 ± 0.28; $n = 3$ animals; *Figure 5G, H*; *Table 5*; *Figure 5—source data 1*). These and the double retrograde tracing results indicate that mPFC$_{NAc}$ and mPFC$_{VTA}$ neurons are rather nonoverlapping, although there is a marginal population – in the L5 – that innervates both areas.

**Table 5.** Quantification of axon length and density in the ventral tegmental area (VTA) and nucleus accumbens (NAc) in the mPFC$_{NAc}$, mPFC$_{VTA}$, mPFC$_{Rbp4}$, and mPFC$_{Thy1}$ animals ($n = 3$ mice in each group).

| | | NAc | VTA | | | NAc | VTA |
|---|---|---|---|---|---|---|---|
| mPFC$_{NAc}$ | Axon length (mm) | 907.58 | 30.80 | PFC$_{Rbp4}$ | Axon length (mm) | 597.27 | 299.93 |
| | | 1385.42 | 24.23 | | | 617.26 | 353.48 |
| | | 401.99 | 9.70 | | | 1323.86 | 99.42 |
| | Volume (mm³) | 0.0068 | 0.018 | | Volume (mm³) | 0.0069 | 0.019 |
| | | 0.0060 | 0.020 | | | 0.0059 | 0.0089 |
| | | 0.0028 | 0.019 | | | 0.0032 | 0.016 |
| | Density (mm/mm³) | 49,215.43 | 4539.63 | | Density (mm/mm³) | 31,674.39 | 43,542.44 |
| | | 69,073.73 | 4042.84 | | | 69,696.59 | 60,044.90 |
| | | 20,991.09 | 3519.42 | | | 80,501.53 | 31,307.55 |
| | Relative density* | 0.092 | | | Relative density* | 1.37 | |
| | | 0.059 | | | | 0.86 | |
| | | 0.17 | | | | 0.39 | |
| mPFC$_{VTA}$ | Axon length (mm) | 130.72 | 196.69 | mPFC$_{Thy1}$ | Axon length (mm) | 1708.58 | 517.52 |
| | | 129.94 | 161.77 | | | 2719.73 | 459.37 |
| | | 46.42 | 63.91 | | | 2807.45 | 398.95 |
| | Volume (mm³) | 0.0067 | 0.017 | | Volume (mm³) | 0.0058 | 0.016 |
| | | 0.0061 | 0.016 | | | 0.0068 | 0.016 |
| | | 0.0064 | 0.014 | | | 0.0057 | 0.013 |
| | Density (mm/mm³) | 7578.49 | 29,542.50 | | Density (mm/mm³) | 104,685.07 | 88,715.34 |
| | | 7954.40 | 26,648.57 | | | 166,781.92 | 67,511.08 |
| | | 3206.06 | 9953.67 | | | 210,241.59 | 69,900.41 |
| | Relative density* | 3.90 | | | Relative density* | 0.85 | |
| | | 3.35 | | | | 0.40 | |
| | | 3.10 | | | | 0.33 | |

*Relative density = RAD$_{VTA}$/RAD$_{NAc}$.

## mPFC$_{NAc}$ and mPFC$_{VTA}$ populations have different efferent connections

After confirming that mPFC$_{NAc}$ and mPFC$_{VTA}$ neurons are mostly separated at the cellular level, we sought to investigate the projection pattern of these populations throughout the brain. Therefore, we used immunoperoxidase development with DAB-Ni as a chromogen (IHC$_{DAB-Ni}$) (*Figure 3—figure supplement 1*) for the mPFC$_{NAc}$ (n = 3 mice) and mPFC$_{VTA}$ (n = 3 mice) brain samples of *CAV2-Cre*-mediated viral labeling. Semi-quantitative investigation of the samples revealed clear differences between the two populations (*Figure 6*; *Table 6*). Most notably, mPFC$_{NAc}$ neurons projected intensively to the ipsi- and contralateral striatum – including the NAc (*Figure 6C*, left) –, various cortical areas (*Figure 6A–H*, left), and the amygdala (*Figure 6F*, left). On the other hand, mPFC$_{VTA}$ innervation was strongest in the lateral (LS) and medial septum (MS; *Figure 6C*, right), the hypothalamus (HT), the bed nucleus of the stria terminalis (BNST; *Figure 6D–F*, right), the midline thalamic nuclei (*Figure 6E, F*), the zona incerta (ZI; *Figure 6F*, right) and various tectal (*Figure 6G, H*, right), tegmental – including the VTA – (*Figure 6G–I*, right) and pontine regions (*Figure 6G–I*, right). Taken together, our investigation revealed that mPFC$_{NAc}$ and mPFC$_{VTA}$ populations differ in their projection patterns not only in the NAc and VTA, but throughout the brain.

## Discussion

Here, we described the molecular, neurochemical, and anatomical characteristics of mPFC regions and layers. Relying on this framework, we found that most mPFC neurons projecting to the NAc and the VTA were distributed in the same subregions, although with varying densities. Furthermore, these populations were mainly located in different layers (*Figure 7*). Accordingly, mPFC$_{NAc}$ and mPFC$_{VTA}$ neuron populations showed minimal overlap at the cellular level, expressed different combination of layer-specific molecular markers and their efferent connections showed clear differences throughout the brain. While mPFC$_{NAc}$ neurons mostly innervated ipsi- and contralateral cortical, striatal, and amygdalar regions, mPFC$_{VTA}$ axons were most abundant in various ipsilateral diencephalic and mesencephalic areas.

Generally, mPFC$_{NAc}$ and mPFC$_{VTA}$ neurons were found in the same subregions, namely the PrL, MO, IL, Cg1, DP, and DTT, confirming previous results (*Gabbott et al., 2005*). However, one notable difference emerged between the two populations. While mPFC$_{NAc}$ neurons formed one, mostly continuous cluster with the highest number of cells in the PL and MO, mPFC$_{VTA}$ neurons formed two visually distinct laminar clusters: one in the middle and another in the deeper part of the mPFC.

Regarding their laminar distribution, mPFC$_{NAc}$ neurons were mostly (~90%) found in the superficial layers (L2/3 and L5a), as previously reported (*Kim et al., 2017*). Traditionally, most striatum-projecting cortical neurons belong to the IT projection group (*Harris and Shepherd, 2015*). High ratio of Calb1-expressing neurons in the L2/3 (~70%) and strong innervation of the NAc in the *Calb1-Cre* animals also suggest their IT-like nature, since Calb1 is considered to be an IT marker (*Harris et al., 2019*). The functional importance of these L2/3 mPFC cells has been shown by *Shrestha et al., 2015* demonstrating that their genetic perturbation leads to augmented depressive behavior in response to stressful events, possibly via the NAc–hypothalamic pathway.

In addition to Calb1, Rbp4 – a genetic marker for both IT and PT neurons (*Rojas-Piloni et al., 2017*; *Harris et al., 2019*) – was also expressed to some extent in the L2/3 besides the L5 of the mPFC. Accordingly, these cells provided strong input to NAc. Surprisingly, despite their relatively low number, mPFC neurons expressing Ntsr1, distributed only in the L5a, also heavily innervated the NAc. These observations indicate regional differences in the distribution of the Rbp4- and Ntsr1-expressing cortical neurons, since Rbp4 is known to be present in the L5, while Ntsr1 is a generally used marker for L6 CT neurons in other, mostly primary cortical regions (*Jeong et al., 2016*; *Sundberg et al., 2018*; *Matho et al., 2021*). We confirmed these results using the same viral tracing experimental approach and the same animal strains targeting the neighboring primary motor cortex to exclude the possibility of a faulty mouse/viral strain. In fact, Rbp4 cells were exclusively localized in the L5 of M1. Furthermore, Ntsr1 neurons were only distributed in the L6 of the M1 and innervated the thalamus but

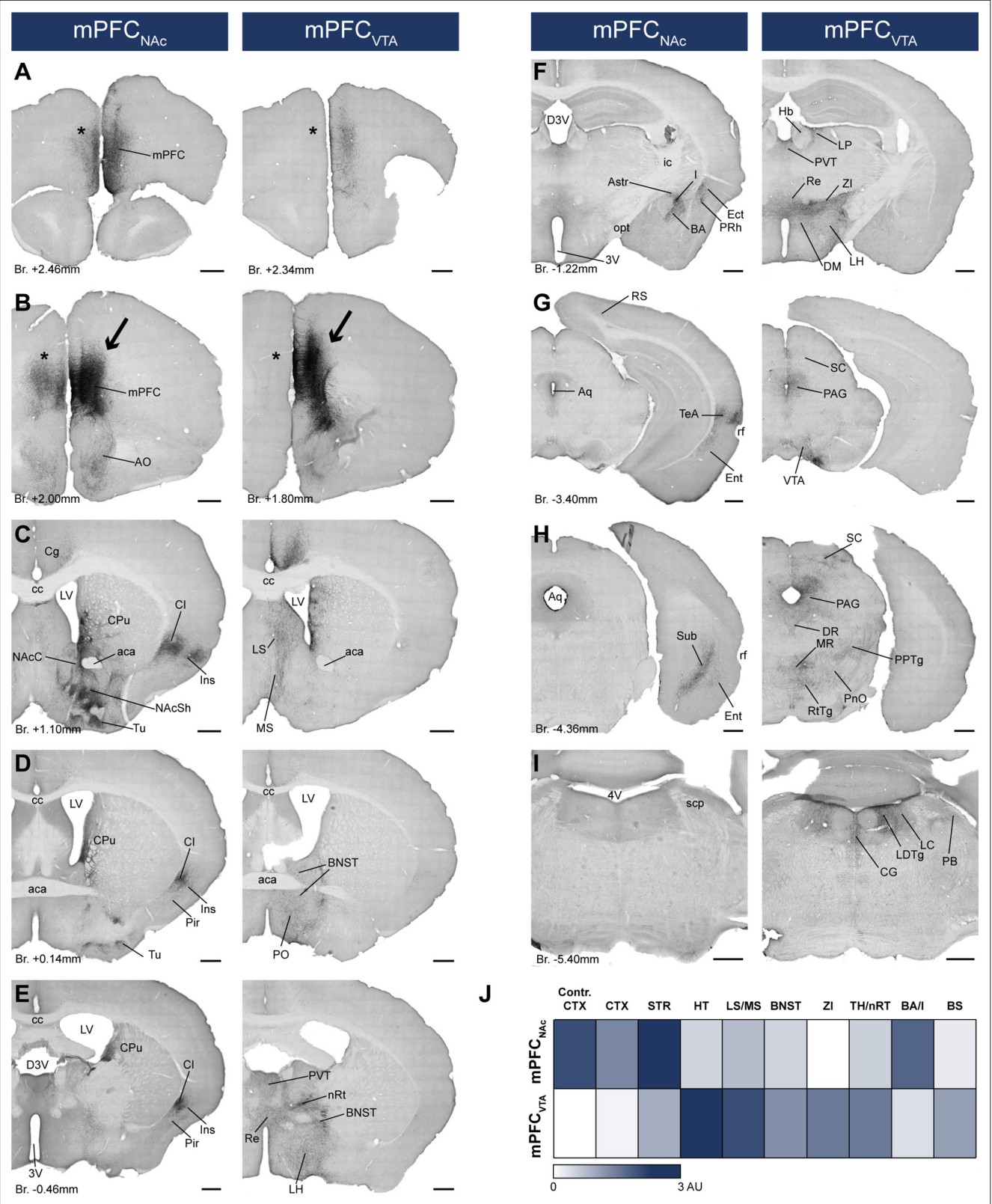

**Figure 6.** mPFC_NAc and mPFC_VTA neurons possess different efferent connections. (**A–I**) Brightfield images showing the distribution of *CAV2-Cre*-mediated AAV-DIO-mCherry-labeled axons visualized with IHC_DAB-Ni at different AP levels. Arrows indicate adeno-associated viral vector (AAV) injection sites in the medial prefrontal cortex (mPFC) (**B**). Note the clear difference between the mPFC_NAc (left column) and mPFC_VTA (right column) populations, most prominently in the striatum (**C**), different cortical areas (**A–H**), the hypothalamus (**D–F**), and the brainstem (**F–I**) – including the ventral tegmental area

*Figure 6 continued on next page*

*Figure 6 continued*

(VTA) (**G**). Note the almost complete lack of contralateral cortical projection in mPFC$_{VTA}$ animals as opposed to mPFC$_{NAc}$ animals (**A, B**, asterisks). For experimental design see *Figure 6A*. (**J**) Summary table showing the innervation intensities of mPFC$_{NAc}$ (top row) and mPFC$_{VTA}$ (bottom row) populations (*n* = 3–3 mice) in different brain regions. Darker color indicates stronger innervation. For details see *Table 6*. Scale bars: 500 µm. 3V, 3rd ventricle; 4V, 4th ventricle; aca, anterior commissure, anterior part; AO, anterior olfactory nucleus; Astr, amygdalostriatal transition area; Aq, aqueduct; BA, basolateral amygdaloid nucleus; BNST, bed nucleus of the stria terminalis; BS, brainstem; cc, corpus callosum; CG, central gray; CPu, caudate putamen; CTX, cortex; D3V, dorsal 3rd ventricle; DM, dorsomedial hypothalamic nucleus; DR, dorsal raphe; Ect, ectorhinal cortex; Ent, entorhinal cortex; Hb, habenula; I, intercalated amygdalar nuclei; ic, internal capsule; Ins, insular cortex; LC, locus coeruleus; LDTg, laterodorsal tegmental nucleus; LP, lateral posterior thalamic nucleus; LS, lateral septum; LV, lateral ventricle; MR, medial raphe; MS, medial septum; nRT, reticular thalamic nucleus; PVT, paraventricular thalamic nucleus; VDB, nucleus of the vertical limb of the diagonal band; VP, ventral pallidum; opt, optic tract; PAG, periaqueductal gray; PB, parabrachial nucleus; PnO, pontine reticular nucleus, oral part; PO, preoptic area; PPTg, pedunculopontine tegmental nucleus; PRh, perirhinal cortex; Re, reuniens thalamic nucleus; rf, rhinal fissure; RS, retrosplenial cortex; RtTg, reticulotegmental nucleus of the pons; SC, superior colliculus; scp, superior cerebellar peduncle; STR, striatum; Sub, subiculum; TeA, termporal association cortex; TH, thalamus; Tu, olfactory tubercle; ZI, zona incerta.

not the striatum (data not shown). These results indicate that some molecular markers have distinct laminar distribution and projection patterns in primary and higher-order cortical areas.

Further supporting this notion, we demonstrated that Ctip2, which is generally present in PT neurons of the L5b–L6 (*Arlotta et al., 2005*; *Ueta et al., 2014*; *Kim et al., 2017*) was expressed in about one-fifth of all mPFC$_{NAc}$ (IT-like) neurons. This suggests that either some PT-like mPFC neurons innervate the striatum or, alternatively, some IT-like neurons express Ctip2 in the mPFC. Previous results reported that PT neurons can innervate the striatum (*Economo et al., 2018*; *Matho et al., 2021*; *Gao et al., 2022*) supporting the first option. However, to the best of our knowledge, there is no direct evidence for the complete absence of Ctip2 expression in IT neurons, so we cannot completely rule out the second possibility either.

While mPFC$_{NAc}$ neurons were present rather superficially, mPFC$_{VTA}$ neurons were mostly (~95%) localized in the deeper layers, namely in L5b and L6 (*Geisler and Zahm, 2005*) and the vast majority (~95%) of them expressed Ctip2. Furthermore, Rbp4 neurons – shown to have a reinforcing effect (*Pan et al., 2021*) – innervated the VTA and the NAc with similar intensity. If we assume that IT- and PT-like Rbp4 neurons are spatially separated (in L2/3–L5a and L5b, respectively), and that IT-like neurons innervate the NAc but not the VTA, then, these results suggest that mPFC$_{VTA}$ neurons have a PT-like phenotype. However, FoxP2, a L6 CT neuron marker (*Kast et al., 2019*; *Matho et al., 2021*) was also expressed by almost half of all mPFC$_{VTA}$ cells. This observation was confirmed by cell-specific viral tracing in the *FoxP2-Cre* mouse strain, where labeled neurons were found in the L6 – and to some extent in the L5 – and projected heavily to the VTA and to the thalamus (data not shown), resembling a mixed PT–CT population. Accordingly, axons of the *CAV2-Cre*-labeled mPFC$_{VTA}$ neurons collateralized to the thalamus as well. In contrast, FoxP2 neurons in the M1 cortex showed clear CT phenotype (data not shown), as it was previously reported (*Matho et al., 2021*). These results strengthened our previous assumption that some cell types have different anatomical phenotype in primary and prefrontal cortical regions.

The different laminar distribution and molecular characteristics of mPFC$_{NAc}$ and mPFC$_{VTA}$ neurons suggest that these populations are mostly separated. However, previous publications yielded contradictory results about the target selectivity of mPFC neurons, which can be resolved, if we consider that multiple projection was found to be high when the experiments were carried out in one neuron population (e.g., only IT or only PT neurons) (*Thierry et al., 1983*; *Ferino et al., 1987*; *Cassell et al., 1989*; *Vázquez-Borsetti et al., 2011*; *Rojas-Piloni et al., 2017*), but low when the experiments involved mixed populations (e.g., PT and IT neurons) (*Pinto and Sesack, 2000*; *Gabbott et al., 2005*; *Morishima and Kawaguchi, 2006*). Accordingly, in most studies addressing this question, NAc- and VTA-projecting (i.e., IT and PT, respectively) populations were described as separate (*Pinto and Sesack, 2000*; *Kim et al., 2017*; *Cruz et al., 2021*) in good accordance with our results. In contrast, *Gao et al., 2020* found relatively high overlap between NAc- and VTA-projecting neurons in the Cg1.

**Table 6.** Whole-brain mapping data showing the axon densities in different brain regions in the mPFC$_{NAc}$ and mPFC$_{VTA}$ animals ($n$ = 3–3 mice).

| | | mPFC$_{VTA}$ | | | mPFC$_{NAc}$ | | |
|---|---|---|---|---|---|---|---|
| | | mPFC$_{VTA}$ #1 | mPFC$_{VTA}$ #2 | mPFC$_{VTA}$ #3 | mPFC$_{NAc}$ #1 | mPFC$_{NAc}$ #2 | mPFC$_{NAc}$ #3 |
| | Contralat. mPFC | | | | ++ | +++ | +++ |
| | Ins/Cl | + | | + | +++ | ++ | +++ |
| | RS | + | | + | + | + | + |
| | TeA | | | | ++ | ++ | +++ |
| | Pir | | | | + | + | + |
| | Ect | | | | ++ | + | ++ |
| | Sub | | | | ++ | ++ | +++ |
| CTX | Ent | | | | + | ++ | ++ |
| | NAc | + | + | + | +++ | +++ | +++ |
| | CPu | ++ | + | ++ | +++ | +++ | +++ |
| STR | Tu | + | + | + | +++ | +++ | +++ |
| | PVT | +++ | +++ | +++ | ++ | + | + |
| | Re | ++ | +++ | +++ | + | + | + |
| | LP | ++ | + | ++ | + | | |
| | DLG | | | + | | | |
| | PIL | + | + | + | | | |
| TH | nRT | ++ | ++ | +++ | + | + | + |
| | MeA | ++ | + | ++ | + | | + |
| | Astr | | | | ++ | ++ | +++ |
| | CeA | + | | + | + | | + |
| | I | | | + | ++ | ++ | ++ |
| AMY | BA | + | | + | +++ | ++ | +++ |

*Table 6 continued on next page*

*Table 6 continued*

| | | mPFC_VTA | | | mPFC_NAc | | |
|---|---|---|---|---|---|---|---|
| | pv | +++ | ++ | +++ | + | + | + |
| | PAG | +++ | ++ | +++ | + | + | + |
| | VTA | +++ | +++ | +++ | + | + | + |
| | SC | + | | ++ | | | |
| | SN | ++ | + | + | + | | |
| | MR | ++ | ++ | + | | | |
| | DR | ++ | + | ++ | + | + | |
| | DpMe | + | | + | | | |
| | IC | + | | + | | | |
| | PPTg | + | | + | | | |
| | PnO | ++ | + | +++ | | | |
| | RR | + | | + | + | | |
| | RtTg | + | + | ++ | | | + |
| | LDTg | ++ | + | +++ | + | + | + |
| | PB | ++ | ++ | ++ | | | |
| | CG | ++ | + | +++ | + | + | + |
| | LC | + | ++ | + | | + | |
| | DMTg | | | ++ | | | |
| | Pn | | | + | | | |
| BS | SubB | + | | ++ | | | |
| | VP | ++ | ++ | ++ | + | + | + |
| | BNST | ++ | + | ++ | + | | + |
| | Septum | +++ | ++ | +++ | + | + | + |
| | HT | +++ | +++ | +++ | + | | + |
| | NB/SI | + | + | + | + | + | ++ |
| | Hb | + | | + | + | | |
| Others | ZI | ++ | + | +++ | | | |

AMY = amygdala. Astr = amygdalostriatal transition area. BA = basolateral amygdaloid nucleus. BNST = bed nucleus of the stria terminalis. BS = brainstem. CeA = central amygdaloid nucleus. CG = central gray. Cl = claustrum. CPu = caudate putamen. CTX = cortex. DLG = dorsal lateral geniculate nucleus. DMTg = dorsomedial tegmental area. DpMe = deep mesencephalic nucleus. DR = dorsal raphe. Ect = ectorhinal cortex. Ent = entorhinal cortex. Hb = habenula. HT = hypothalamus. I = intercalated amygdalar nuclei. IC = inferior colliculus. Ins = insular cortex. LC = locus coeruleus. LDTg = laterodorsal tegmental nucleus. LP = lateral posterior thalamic nucleus. MeA = medial amygdaloid nucleus. MR = medial raphe. NB = basal nucleus. nRT = reticular thalamic nucleus. PVT = paraventricular thalamic nucleus. VP = ventral pallidum. PAG = periaqueductal gray. PB = parabrachial nucleus. PIL = posterior intralaminar thalamic nucleus. Pir = piriform cortex. Pn = pontine nuclei. PnO = pontine reticular nucleus, oral part. PPTg = pedunculopontine tegmental nucleus. pv = periventricular fiber system. Re = reuniens thalamic nucleus. RR = retrorubral nucleus. RS = retrosplenial cortex. RtTg = reticulotegmental nucleus of the pons. SC = superior colliculus. SI = substantia innominata. STR = striatum. Sub = subiculum. SubB = subbrachial nucleus. TeA = termporal association cortex. TH = thalamus. Tu = olfactory tubercule. ZI = zona incerta.

Here, we described the Cg1 as a minor source of input for both the NAc (<3%) and the VTA (<10%) compared to other mPFC subregions, which might provide explanation for this contradiction.

A recent publication (*Gao et al., 2022*) investigating fully reconstructed mPFC neurons demonstrated that IT and PT neurons collateralize extensively, although this collateralization was strongest for traditional target regions of IT and PT classes. This notion is further supported by similar experiments carried out in the M1 cortex (*Callaway, 2021*; *Muñoz-Castañeda et al., 2021*; *Peng et al., 2021*). We also investigated the multiple-projecting nature of mPFC neurons in the mesocortico-limbic system and found that retrogradely labeled mPFC_NAc and mPFC_VTA neurons showed minimal overlap (<2%), indeed. Furthermore, using *CAV2-Cre*-mediated viral tracing we demonstrated that

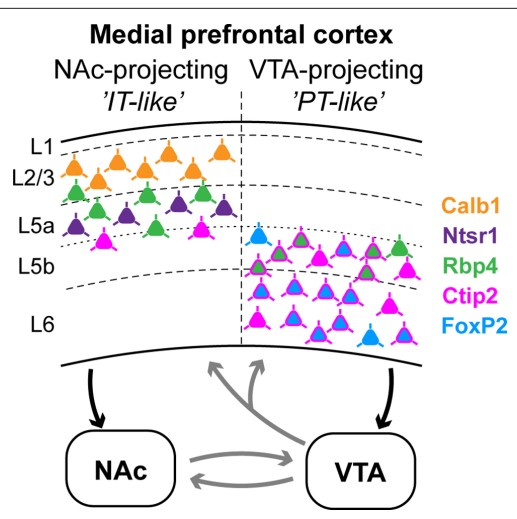

**Figure 7.** Summary: molecular characteristics and laminar distribution of the two identified projection groups in the medial prefrontal cortex (mPFC). Neurons that innervate the nucleus accumbens (NAc) ('IT-like') are mostly localized in the upper layers of the mPFC (L2/3–5a) and express Calb1 (green), Ntsr1 (purple), Rbp4 (orange), and to a lesser extent, Ctip2 (magenta). mPFC cells that innervate the ventral tegmental area (VTA) ('PT-like') are mostly localized in the deeper layers (L5b–6) and express Ctip2, FoxP2 (cyan), and Rbp4. Connections between NAc and VTA, and ascending VTA pathways (gray arrows) are based on literature data (see Introduction, Discussion).

mPFC$_{NAc}$ cells innervate the NAc approximately 10 times stronger than the VTA. On the other hand, mPFC$_{VTA}$ neurons also showed clear preference (3.5-fold) for the VTA over the NAc. Considering that mPFC innervates the VTA with a relatively sparse axon arborization (**Carr and Sesack, 2000**; **Geisler and Zahm, 2005**), these result further support that these populations are rather nonoverlapping at the single-cell level. However, complete projection pattern analysis revealed that neurons of these populations collateralize extensively to innervate different areas throughout the brain, in accordance with the findings of **Gao et al., 2022**. Specifically, mPFC$_{NAc}$ neurons showed IT-like projection pattern (mainly ipsi- and contralateral cortical, amygdalar, and striatal targets), while mPFC$_{VTA}$ efferents resembled PT neurons (mainly ipsilateral mesencephalic and diencephalic targets).

In general, mPFC$_{NAc}$ neurons participate in a range of reward-related tasks. For example, activation of mPFC$_{NAc}$ neurons suppresses reward seeking in a conflicting situation (**Kim et al., 2017**). On the other hand, others reported that optical stimulation of mPFC$_{NAc}$ neurons promote conditioned reward seeking (**Otis et al., 2017**). In accordance, **Britt et al., 2012** demonstrated that optical stimulation of mPFC terminals in the NAc can facilitate self-stimulation, although **Stuber et al., 2011** reported the lack of such effect. Therefore, it seems plausible that there is a topographical segregation within the mPFC-to-NAc pathway with different functional properties or different cell types convey different behavioral information, or the combination of both. Similarly, it was previously reported that mPFC neurons can excite and inhibit VTA dopamine neurons equally (**Lodge, 2011**), which also suggests functional separation within the mesocorticolimbic system. Recent findings of topographically biased input–output connectivity of different mPFC (**Cruz et al., 2021**) and VTA dopamine neurons (**Aransay et al., 2015**; **Beier et al., 2015**), as well as high topographic precision in corticostriatal pathways (**Hooks et al., 2018**) further support this suggestion. So, cell-specific studies are needed to completely clarify the functional complexity of these pathways.

Taken together, mPFC$_{NAc}$ and mPFC$_{VTA}$ populations are rather nonoverlapping and their afferent connectivity shows IT- and PT-like features, respectively. However, high CT marker (FoxP2) expression in mPFC$_{VTA}$ neurons, as well as PT (Ctip2) and CT (Ntsr1) marker expression in mPFC$_{NAc}$ neurons indicate that the traditional IT–PT–CT classes might have somewhat different molecular characteristics in mPFC compared to the well-studied primary cortical areas. In accordance, a recent publication also demonstrated high genetic diversity of mPFC neurons (**Gao et al., 2022**), even within projection neuron classes. Therefore, in the future, understanding the versatility of prefrontal cortical influence over mesocorticolimbic functions requires a combination of molecular-, cellular-, laminar-, and region-specific approaches.

## Anatomical considerations

It is generally accepted that the rodent mPFC is anatomically homologous to the primate anterior cingulate cortex (**Russo and Nestler, 2013**; **Vogt and Paxinos, 2014**). However, there are notable nomenclatural inconsistencies (**Laubach et al., 2018**; **Le Merre et al., 2021**) in the rodent mPFC literature (**Lodge, 2011**; **Bossert et al., 2012**; **Adhikari et al., 2015**; **Shrestha et al., 2015**; **Warren**

*et al., 2019*; *Lichtenberg et al., 2021*). For example, the exact definition of the PrL subregions greatly varies between publications, just like the distinction between dorsal and ventral mPFC. Such inaccuracies can contribute to the still abundant contradictions in the literature and complicate the proper interpretation of the results.

To overcome these setbacks, we combined multiple IHC$_{Fluo}$ against different molecular markers that can (1) delineate the borders between different subregions (PV, Calb1) (*van Brederode et al., 1991*; *Sun et al., 2002*; *Akhter et al., 2014*; *Mátyás et al., 2014*) and (2) clearly define cortical layers in the mPFC (Calb1, Ctip2, and FoxP2) (*Ferland et al., 2003*; *Kim et al., 2017*). We always used these markers to locate injection sites and labeled neurons within the mPFC. Reliable primary antibodies raised in several different species against all of these markers are commercially available and they can be combined easily. Therefore, we suggest the general adoption of this method to precisely define and separate mPFC subregions and layers in future studies.

## Materials and methods
### Animals
Adult (3–5 months old, male and female; $n_{total}$ = 38) wild-type ($n$ = 21; $n_{male}$ = 9; $n_{female}$ = 12), *Rbp4-Cre* (*Tg(Rbp4-cre)KL100Gsat*, RRID: MMRRC_037128-UCD, gift from L. Acsády; $n$ = 4; $n_{female}$ = 4), *Thy1-Cre* (*FVB/N-Tg(Thy1-cre)1Vln/J*, RRID: IMSR_JAX:006143; gift from B. Rózsa; $n$ = 3; $n_{male}$ = 1; $n_{female}$ = 2), *Calb1-Cre* (*B6;129S-Calb1$^{tm2.1(cre)Hze}$/J*, RRID: IMSR_JAX:028532; $n$ = 3; $n_{male}$ = 2; $n_{female}$ = 1), *Ntsr1-Cre* (*Tg(Ntsr1-cre)GN220Gsat*, RRID: MMRRC_017266-UCD a gift from P. Barthó; $n$ = 3; $n_{male}$ = 3), and *FoxP2-Cre* mice (*B6.Cg-Foxp2$^{tm1.1(cre)Rpa}$/J*, RRID: IMSR_JAX:030541; $n$ = 3; $n_{male}$ = 2; $n_{female}$ = 1) were used for the experiments. Animals were group housed in a humidity- and temperature-controlled environment. Animals were entrained to a 12 hr light/dark cycle (light phase from 07:00 AM) with food and water available ad libitum. All procedures were approved by the Regional and Institutional Committee of the Research Centre for Natural Sciences and the Institute of Experimental Medicine. The experiments were approved by the National Animal Research Authorities of Hungary (PEI/001/2290-11/2015).

### Stereotactic surgeries
#### Classical retrograde tracing
All animals were anesthetized under ketamine–xylazine (5:1, 3× dilution, ketamine: 100 mg/kg; xylazine: 4 mg/kg) during all anatomical surgeries. Single and double retrograde tracing surgeries were carried out with 0.5% CTB subunit (List Biological Laboratories: 104) and/or 2% FG (Fluorochrome LLC) to reveal the prefrontal cortical source of NAc (AP/L/DV: +1.4/±0.8/3.9–4.2) and VTA (AP/L/DV: −3.3/±0.3/4.0–4.2) innervation. Tracers were iontophoretically injected (7–7 s on/off duty cycle, 3–5 μA, for 5–10 min) with IonFlow Bipolar electrophoretic equipment (Supertech Instruments Hungary). After all surgeries, animals received Rimadyl (Carprofen, 1.4 mg/kg).

For anatomical analysis, after 7 days of survival time, mice were perfused transcardially first with saline (~50 ml), then, with ~150 ml of fixative solution containing 4% paraformaldehyde (Sigma-Aldrich, CAS No. 30525-89-4) in 0.1 M phosphate buffer (PB).

#### Exclusion criteria
Animals in which the injections sites or tracer tracks reached regions that could affect labeling (e.g., caudate putamen, substantia nigra, ventral pallidum) were excluded from further analysis. A total of $n$ = 8 animals were excluded.

#### Identification of different brain regions and cortical layers
We used different neurochemical markers to identify brain regions of interest and to separate cortical layers in the tissue samples labeled with fluorescent immunohistochemistry (IHC$_{Fluo}$). Calbindin (Calb1) staining (see below) was used to delineate the core (strong Calb1 expression) and shell (weak Calb1 expression) region of the NAc (*Jongen-Rêlo et al., 1994*), and TH staining for the VTA (*Oades and Halliday, 1987*; *Morales and Margolis, 2017*). Layer 2/3 (L2/3) of the cerebral cortex was identified using Calb1 staining (*van Brederode et al., 1991*; *Sun et al., 2002*), while L6 with forkhead box

protein P2 (FoxP2) staining (*Ferland et al., 2003*). COUP-TF-interacting protein 2 (Ctip2) staining was used to label L5b and L6 (*DeNardo et al., 2015*; *Figure 1—figure supplement 1*).

We used the 2nd Edition of the Mouse Brain is Stereotaxic Coordinates by *Paxinos and Franklin, 2001* as a reference, because the vast majority of mPFC literature uses this nomenclature. In comparison with the newest, 5th edition (*Franklin and Paxinos, 2019*), the mPFC region we defined as prelimbic cortex (PrL) is approximately equivalent to the A32 area, the IL to the A25, and the rostral aspects of the cingulate cortex, area 1 and 2 (Cg1–2) to the A24b and A24a, respectively. The secondary motor (M2), MO, DP, and DTT regions have not changed significantly between the two editions.

## Anterograde viral tracing

For cell type-specific anterograde viral tracing AAV5.EF1a.DIO.eYFP.WPRE.hGH (30–100 nl; Penn Vector Core; #27056-AAV5; titer: $5 \times 10^{12}$ GC/ml) or AAV5-EF1a-DIO-mCherry viruses (30–100 nl; UNC Vector Core; #50462; titer: $7 \times 10^{12}$ GC/ml) were injected at a rate of 0.5–1 nl/s into mPFC (AP/L/DV: +1.7–1.9/±0.3/2.1–1.6 mm) and M1 (AP/L/DV: +1.4/±1.6/1.3–1.0 mm) using a Nanoliter Injector (World Precision Instruments, FL, USA).

Animals were perfused (see above) after 4–6 weeks of survival time. Viral expression was always analyzed after IHC$_{Fluo}$ enhancement (*Figure 3—figure supplement 1*; *Falcy et al., 2020*), even for eYFP (see below).

## Intersectional retro-anterograde viral tracing

In order to selectively label NAc- (mPFC$_{NAc}$) and VTA-projecting mPFC cells (mPFC$_{VTA}$), we injected *Canine adenovirus type 2 carrying Cre-recombinase gene* (*CAV2-Cre*, CMV promoter, titer: $2.5 \times 10^{10}$ pp/ml, Plateforme de Vectorologie de Montpellier, France; a gift from D. Zelena) into the NAc (*n* = 3 animals) or VTA (*n* = 3 animals) (see coordinates above) of wild-type animals, mixed with 5% biotinylated dextrane amine (BDA, MW: 10.000, Molecular Probes: D1956, RRID: AB_2307337; 1:1; 80–120 nl/animal; 1 nl/s). Note that BDA was used to locate the tip of the injecting pipette (*Figure 5B1, C1*), not the whole extent of viral diffusion. At the same time, the mPFC (see coordinates above) of the same animals was injected with AAV5-EF1a-DIO-mCherry (see details above). After 6 weeks of survival, animals were perfused, and their brains were processed for further analysis (see above).

## Tissue processing and immunohistochemistry

Tissue blocks were cut on a VT1200S Vibratome (Leica) into 50 µm coronal sections. Free-floating sections were intensively washed with 0.1 M PB. All antibodies were diluted in 0.1 M PB. For fluorescent labeling, sections were first treated with a blocking solution containing 10% normal donkey serum (NDS, Sigma-Aldrich: S30-M) or 10% normal goat serum (NGS, Vector: S-1000, RRID: AB_2336615) and 0.5% Triton-X (Sigma-Aldrich, CAS Number: 9036-19-5) in 0.1 M PB for 30 min at room temperature (RT).

## Fluorescent immunohistochemistry

Sections were incubated in primary antibody solution overnight at RT or for 2–3 days at 4°C. The following primary antibodies were used: green fluorescent protein (GFP, chicken, Life Technology: A10262, RRID: AB_2534023; 1:2000), mCherry (mCherry; rabbit, BioVision: 5993-100, RRID: AB_1975001; 1:2000), red fluorescent protein (RFP; rat, Chromotek: 5F8, RRID: AB_2336064; 1:2000), FoxP2 (mouse, Merck Millipore: MABE415, RRID: AB_2721039; 1:2000; Invitrogen: MA5-31419, RRID: AB_2787055; 1:2000; rabbit, Abcam: ab16046, RRID: AB_2107107; 1:500), Calb1 (rabbit, SWANT: CB38, RRID: AB_10000340; 1:2000; mouse, SWANT: 300, RRID: AB_10000347; 1:2000; chicken, Synaptic Systems: 214 006, RRID: AB_2619903; 1:2000), TH (mouse, Immunostar: 22941, RRID: AB_572268; 1:8000), FG (rabbit, FluoroChrome, 1:50.000; guinea pig, Protos Biotech: NM-101, RRID: AB_2314409; 1:5000), CTB (goat, List Biological Laboratories: 703; 1:20.000), PV (mouse, SWANT: PV 235, RRID: AB_10000343; 1:2000), and Ctip2 (rat, Abcam: ab18465, RRID: AB_2064130; 1:500).

For IHC$_{Fluo}$ staining, after primary antibody incubation, sections were treated with the following secondary IgGs (1:500; 2 hr at RT): Alexa 488-conjugated donkey anti-rabbit (DAR-A488; Jackson: 711-545-152, RRID: AB_2313584), donkey anti-mouse (Jackson: 715-545-150, RRID: AB_2340846), goat anti-chicken (Molecular Probes: A11039, RRID: AB_142924), donkey anti-guinea pig (Jackson: 706-545-148, RRID: AB_2340472); Alexa 555-conjugated donkey anti-goat (Molecular Probes: A21432,

RRID: AB_141788), donkey anti-mouse (Molecular Probes: A31570, RRID: AB_2536180), donkey anti-rat (Southern Biotech: 6430-32, RRID: AB_2796359); Cy3-conjugated donkey anti-rabbit (Jackson: 715-165-152, RRID: AB_2307443), donkey anti-mouse (Jackson: 715-165-151, RRID: AB_2340813); Alexa 594-conjugated donkey anti-mouse (Molecular Probes: A21203, RRID: AB_141633), donkey anti-rabbit (Molecular Probes: A21207, RRID: AB_141637), Alexa 647-conjugated donkey anti-mouse (Jackson: 715-605-151, RRID: AB_2340863; Invitrogen: A-31571, RRID: AB_162542), or donkey anti-rabbit (Jackson: 711-605-152, RRID: AB_2492288).

When necessary, staining was enhanced after primary antibody incubation with biotinylated secondary antibodies (biotinylated horse anti-goat IgG, Vector Laboratories: BA-9500, RRID: AB_2336123; 1:300; biotinylated goat anti-rabbit – bGAR, Vector Laboratories: BA-1000, RRID: AB_2313606; 1:300; biotinylated goat anti-guinea pig, Vector Laboratories: BA-7000, RRID: AB_2336132; 1:300; 1.5 hr, RT), Elite Avidin-Biotin Complex (eABC, 1:300, Vector Laboratories: PK-6100, RRID: AB_2336819; 1.5 hr, RT), and streptavidin-conjugated fluorescent antibodies (SA-A488, Jackson: 016-540-084, RRID: AB_2337249; 1:2000; SA-Cy3, Jackson: 016-160-084, RRID: AB_2337244; 1:2000; SA-A647, Jackson: 016-600-084, RRID: AB_2341101; 1:2000; 2 hr, RT). All fluorescent slices were mounted in Vectashield (Vector Laboratories: H-1000, RRID: AB_2336789). To reveal the CAV2-Cre/BDA injection site we used eABC (see above) and SA-A488 or SA-A647 (see above).

### Immunoperoxidase staining

For the whole-brain projection pattern analysis of the *CAV2-Cre* animals, we also performed immunoperoxidase staining and used nickel-amplified 3-3′-diaminobenzidine (DAB; Sigma-Aldrich; CAS Number: 91-95-2) technique (DAB-Ni; IHC$_{DAB-Ni}$). Every sixth section (thus, at 300 µm resolution, from Br. + 3.10 to −8.00 mm) was treated first with 1% $H_2O_2$ solution for 10 min, then, after intensive washing, in 10% NDS and 0.2% Triton-X solution as a blocking serum (30 min, RT). After primary antibody incubation (mCherry, see above), slices were incubated in biotinylated secondary antibody (bGAR) and eABC (see above). Then we developed DAB-Ni for 5 min. Sections were then dehydrated in xylol (2 × 10 min) and mounted in DePex (Serva, Heidelberg, Germany; Cat. No. 18243).

### Viral signal amplification

To compare native mCherry expression to IHC$_{Fluo}$ and IHC$_{DAB-Ni}$ enhancement, we stained slices from the *CAV2-Cre* experiments with primary antibody against mCherry and DAR-A488 (see above) (*Figure 3—figure supplement 1A, B*). Then we captured confocal images (see below) from the same brain regions in two channels (i.e., A488 and mCherry). For better visualization, we recolorized the A488 channel at *Figure 3—figure supplement 1A2, B2*. Next, we stained the neighboring slices (i.e., 50 µm apart) with IHC$_{DAB-Ni}$ against mCherry (see above) and captured them with brightfield microscopy (see below) (*Figure 3—figure supplement 1C, D*).

## Microscopy

Fluorescent sections were first analyzed with epifluorescent microscope (Leica DM 2500, Leica Microsystems GmbH; Camera: Olympus DP73, CellSens Entry 1.16, Olympus Corporation) with low magnification (2.5× N PLAN 2.5×/0.07 ∞/-/OFN25, 5× HCX FL PLAN 5×/0.12 ∞/-/B) to find injection sites and labeled cells. Higher magnification (10× Plan Apochromat 10×/0.45 M27; 20× Plan Apochromat 20×/0.8 M27; 63× Plan Apochromat 63×/1.4 Oil DIC M27) images were taken with confocal microscope (Zeiss LSM 710; Zeiss ZEN 2010B SP1 Release version 6.0; Carl Zeiss Microimaging GmbH). Brightfield imaging and whole-brain projection analysis, as well as distribution analysis for retrogradely labeled neurons were completed with a PANORAMIC MIDI II (20× [NA 0.8]; 3DHistech, Hungary) device and the manufacturer's official software (CaseViewer 2.4) for every sixth slice (i.e., at 300 µm resolution).

### Distribution analysis

We used IHC$_{Fluo}$-labeled slices (between Br. + 3.10 to +1.10 mm) to analyze anteroposterior, subregional and laminar distribution of retrogradely labeled mPFC$_{NAc}$ and mPFC$_{VTA}$ neurons ($n$ = 3–3 animals, 5–7 slices/animal). We captured whole slice images at ×20 magnification and manually counted cells using ImageJ (NIH). Note that we simultaneously registered anteroposterior, subregional, and laminar localization of each cell ($N_{mPFCNAc}$ = 1042 neurons; $N_{mPFCVTA}$ = 1878 neurons).

## Colocalization

In order to reveal the proportion of FoxP2-, Ctip2-, and Calb1-positive cells among retrogradely (FG/CTB) labeled $mPFC_{NAc}$ and $mPFC_{VTA}$ cells, we captured ×20 magnification confocal Z-stack (step size: 5 µm) imaging of double-labeled fluorescent sections (3–4 slices/animal, $n$ = 3–3 animals). Labeled cells were then manually analyzed with ImageJ (NIH). Only cells visible in two separate sections with a visible nucleus were analyzed. The same protocol was used to identify double-labeled cells in the double retrograde tracing experiments ($n$ = 3 animals).

## Axon density analysis

We sought to compare mPFC axon densities in the NAc and VTA in the *CAV2-Cre* injected $mPFC_{NAc}$, $mPFC_{VTA}$, and AAV5-EF1a-DIO-mCherry injected *Rbp4*- ($mPFC_{Rbp4}$) and *Thy1-Cre* ($mPFC_{Thy1}$) samples using high-magnification (×63) confocal Z-stacks (step size: 0.27 µm). In the VTA, we captured three stacks in each animal ($n$ = 3 in each strain) at three different AP levels between Bregma −3.10 and −3.80 mm. In the NAc, we captured five–five stacks in the same animals as for the VTA at three different AP levels between Bregma +1.00 and 1.80 mm. We aimed to capture stacks where axon density was visibly the highest at each AP level in each region.

We analyzed the confocal stacks using a custom made automatic ImageJ macro (*Mátyás et al., 2018*) (available at https://github.com/baabek/Axon-density-analyzer-ImageJ-script.git). The macro calculated the axon length for each stack and the total axon length was summed for each brain region in each animal (also see *Figure 5—source data 1*). Then, the total axon length was compared to the summated stack volume (ROI area * number of slices * step size = total volume) for each brain region to calculate the relative axon density (RAD = total axon length/total volume). Then, the ratio of $RAD_{VTA}/RAD_{NAc}$ ($RAD_{(VTA/NAc)}$) was calculated for each animal, where $RAD_{(VTA/NAc)}$ = 1 means that the two areas are equally innervated.

## Statistical analysis

Values are given as mean ± SD. $n$ represents number of animals; $N$ represents cell counts in all figures/tables and their legends. We used SPSS Statistics (ver. 27.0.1.0., IBM) to analyze the axon density data. We used one-way analysis of variance method with least significant difference post hoc test to compare RAD values after testing for the homogeneity of variances. The exact p values are indicated in the figure legends.

No statistical methods were used to predetermine sample size, but it is comparable to previously published work (e.g., *Pinto and Sesack, 2000*; *Faget et al., 2016*).

## Acknowledgements

The authors thank Tamás Herczeg, Réka Erdős, Anna Fehér, and Dóra Zsíros for laboratory assistance, the Institute of Enzymology of the Research Centre for Natural Sciences for the confocal microscope, Dóra Zelena for the CAV2-Cre virus, Norbert Hájos for providing antibodies, László Acsády, Péter Barthó, and Balázs Rózsa for the Cre animals and Csaba Dávid for helping us in axonal analysis. We also thank Norbert Hájos for comments and discussions about the manuscript. This work was supported by the Ministry of Innovation and Technology of Hungary from the National Research, Development and Innovation Fund (FK124434, K138836, and KKP126998 to FM), by the Hungarian Brain Research Program (2017-1.2.1-NKP-2017-00002 to FM), by the Eötvös Loránd Research Network (SA-48/2021 to FM), and by the New National Excellence Program of the Ministry for Innovation and Technology (ÚNKP-21-5-ÁTE-2 to FM; ÚNKP-18-3-II-BME-55, ÚNKP-20-3-II-BME-24, and ÚNKP-21-3-II-BME-61 to ÁB). FM is a János Bolyai Research Fellow.

# Additional information

## Funding

| Funder | Grant reference number | Author |
|---|---|---|
| National Research, Development and Innovation Office | FK124434 | Ferenc Matyas |
| Hungarian Brain Research Program | 2017-1.2.1-NKP-2017-00002 | Ferenc Matyas |
| New National Excellence Program | ÚNKP-21-5-ÁTE-2 | Ferenc Matyas |
| National Research, Development and Innovation Office | K138836 | Ferenc Matyas |
| National Research, Development and Innovation Office | KKP126998 | Ferenc Matyas |
| New National Excellence Program | ÚNKP-18-3-II-BME-55 | Ákos Babiczky |
| New National Excellence Program | ÚNKP-20-3-II-BME-24 | Ákos Babiczky |
| New National Excellence Program | ÚNKP-21-3-II-BME-61 | Ákos Babiczky |
| Eötvös Loránd Research Network | SA-48/2021 | Ferenc Matyas |

The funders had no role in study design, data collection and interpretation, or the decision to submit the work for publication.

## Author contributions

Ákos Babiczky, Conceptualization, Data curation, Software, Formal analysis, Validation, Investigation, Visualization, Methodology, Writing – original draft, Writing – review and editing; Ferenc Matyas, Conceptualization, Resources, Data curation, Supervision, Funding acquisition, Validation, Investigation, Visualization, Methodology, Writing – original draft, Project administration, Writing – review and editing

## Author ORCIDs

Ákos Babiczky (iD) http://orcid.org/0000-0001-5848-7374
Ferenc Matyas (iD) http://orcid.org/0000-0002-3903-8896

## Ethics

All procedures were approved by the Regional and Institutional Committee of the Research Centre for Natural Sciences and the Institute of Experimental Medicine. The experiments were approved by the National Animal Research Authorities of Hungary (PEI/001/2290-11/2015).

## Decision letter and Author response

Decision letter https://doi.org/10.7554/eLife.78813.sa1
Author response https://doi.org/10.7554/eLife.78813.sa2

# Additional files

## Supplementary files
• MDAR checklist

## Data availability

All data generated or analyzed during this study are included in the manuscript and supporting file. All data are presented in figures, tables, figure supplements and source data file.

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

# Appendix 1

## Appendix 1—key resources table

| Reagent type (species) or resource | Designation | Source or reference | Identifiers | Additional information |
|---|---|---|---|---|
| Strain, strain background (*M. musculus*, male, female) | *Tg(Rbp4-cre)KL100Gsat* | MMRRC | RRID: MMRRC_037128-UCD | |
| Strain, strain background (*M. musculus*, male, female) | *FVB/N-Tg(Thy1-cre)1Vln/J* | The Jackson Laboratory | RRID: IMSR_JAX:006143 | |
| Strain, strain background (*M. musculus*, male, female) | *B6;129S-Calb1tm2.1(cre)Hze/J* | The Jackson Laboratory | RRID: IMSR_JAX:028532 | |
| Strain, strain background (*M. musculus*, male, female) | *Tg(Ntsr1-cre)GN220Gsat* | MMRRC | RRID: MMRRC_017266-UCD | |
| Strain, strain background (*M. musculus*, male, female) | *B6.Cg-Foxp2tm1.1(cre)Rpa/J* | The Jackson Laboratory | RRID: IMSR_JAX:030541 | |
| Strain, strain background (*M. musculus*, male, female) | C57BL/6J | The Jackson Laboratory | RRID: IMSR_JAX:000664 | |
| Biological sample (species) | AAV5.EF1a.DIO.eYFP.WPRE.hGH | Penn Vector Core | Cat. #27056-AAV5 | Viral titer: $5 \times 10^{12}$ GC/ml |
| Biological sample (species) | AAV5-EF1a-DIO-mCherry | UNC Vector Core | Cat. #50462 | Viral titer: $7 \times 10^{12}$ GC/ml |
| Biological sample (species) | Canine adenovirus type 2 carrying Cre-recombinase gene | Plateforme de Vectorologie de Montpellier, France | | CMV promoter, titer: $2.5 \times 10^{10}$ pp/ml |
| Biological sample (species) | 10% normal donkey serum | Sigma-Aldrich | S30-M | |
| Biological sample (species) | 10% normal goat serum | Vector | S-1000, RRID: AB_2336615 | |
| Antibody | Anti-GFP (chicken polyclonal) | Life Technology | A10262, RRID: AB_2534023 | 1:2000 |
| Antibody | Anti-mCherry (rabbit polyclonal) | BioVision | 5993-100, RRID: AB_1975001 | 1:2000 |
| Antibody | Anti-RFP (rat monoclonal) | Chromotek | 5F8, RRID: AB_2336064 | 1:2000 |
| Antibody | Anti-FoxP2 (mouse monoclonal) | Merck Millipore | MABE415, RRID: AB_2721039 | 1:2000 |
| Antibody | Anti-FoxP2 (mouse monoclonal) | Invitrogen | MA5-31419, RRID: AB_2787055 | 1:2000 |
| Antibody | Anti-FoxP2 (rabbit polyclonal) | Abcam | ab16046, RRID: AB_2107107 | 1:500 |
| Antibody | Anti-Calb1 (rabbit polyclonal) | Swant | CB38, RRID: AB_10000340 | 1:2000 |
| Antibody | Anti-Calb1 (mouse monoclonal) | Swant | 300, RRID: AB_10000347 | 1:2000 |

*Appendix 1 Continued on next page*

*Appendix 1 Continued*

| Reagent type (species) or resource | Designation | Source or reference | Identifiers | Additional information |
|---|---|---|---|---|
| Antibody | Anti-Calb1 (chicken polyclonal) | Synaptic Systems | 214 006, RRID: AB_261990 | 1:2000 |
| Antibody | Anti-TH (mouse monoclonal) | Immunostar | 22941, RRID: AB_572268 | 1:8000 |
| Antibody | Anti-FG (rabbit polyclonal) | Fluorochrome LLC | | 1:50,000 |
| Antibody | Anti-FG (guinea pig polyclonal) | Protos Biotech | NM-101, RRID: AB_2314409 | 1:5000 |
| Antibody | Anti-CTB (goat, N/A) | List Biological Laboratories | #703 | 1:10,000 |
| Antibody | Anti-PV (mouse monoclonal) | Swant | PV 235, RRID: AB_10000343 | 1:2000 |
| Antibody | Anti-Ctip2 (rat monoclonal) | Abcam | ab18465, RRID: AB_2064130 | 1:500 |
| Antibody | Anti-rabbit, Alexa-488 conjugated (donkey polyclonal) | Jackson | 711-545-152, RRID: AB_2313584 | 1:500 |
| Antibody | Anti-mouse, Alexa-488 conjugated (donkey polyclonal) | Jackson | 715-545-150, RRID: AB_2340846 | 1:500 |
| Antibody | Anti-chicken, Alexa-488 conjugated (goat polyclonal) | Molecular Probes | A11039, RRID: AB_142924 | 1:500 |
| Antibody | Anti-guinea pig, Alexa-488 conjugated (donkey polyclonal) | Jackson | 706-545-148, RRID: AB_2340472 | 1:500 |
| Antibody | Anti-goat, Alexa-555 conjugated (donkey polyclonal) | Molecular Probes | A21432, RRID: AB_141788 | 1:500 |
| Antibody | Anti-mouse, Alexa-555 conjugated (donkey polyclonal) | Molecular Probes | A31570, RRID: AB_2536180 | 1:500 |
| Antibody | Anti-rat, Alexa-555 conjugated (donkey polyclonal) | Southern Biotech | 6430–32, RRID: AB_2796359 | 1:500 |
| Antibody | Anti-mouse, Cy3 conjugated (donkey polyclonal) | Jackson | 715-165-151, RRID: AB_2340813 | 1:500 |
| Antibody | Anti-rabbit, Cy3 conjugated (donkey polyclonal) | Jackson | 715-165-152, RRID: AB_2307443 | 1:500 |
| Antibody | Anti-mouse, Alexa-594 conjugated (donkey polyclonal) | Molecular Probes | A21203, RRID: AB_141633 | 1:500 |
| Antibody | Anti-rabbit, Alexa-594 conjugated (donkey polyclonal) | Molecular Probes | A21207, RRID: AB_141637 | 1:500 |
| Antibody | Anti-mouse, Alexa-647 conjugated (donkey polyclonal) | Jackson | 715-605-151, RRID: AB_2340863 | 1:500 |
| Antibody | Anti-mouse, Alexa-647 conjugated (donkey polyclonal) | Invitrogen | A-31571, RRID: AB_162542 | 1:500 |
| Antibody | Anti-rabbit, Alexa-647 conjugated (donkey polyclonal) | Jackson | 711-605-152, RRID: AB_2492288 | 1:500 |

*Appendix 1 Continued on next page*

*Appendix 1 Continued*

| Reagent type (species) or resource | Designation | Source or reference | Identifiers | Additional information |
|---|---|---|---|---|
| Antibody | Anti-goat, biotinylated (horse polyclonal) | Vector Laboratories | BA-9500, RRID: AB_2336123 | 1:300 |
| Antibody | Anti-rabbit, biotinylated (goat polyclonal) | Vector Laboratories | BA-1000, RRID: AB_2313606 | 1:300 |
| Antibody | Anti-guinea pig, biotinylated (goat polyclonal) | Vector Laboratories | BA-7000, RRID: AB_2336132 | 1:300 |
| Peptide, recombinant protein | Streptavidin, Alexa-488 conjugated | Jackson | 016-540-084, RRID: AB_2337249 | 1:2000 |
| Peptide, recombinant protein | Streptavidin, Cy3 conjugated | Jackson | 016-160-084, RRID: AB_2337244 | 1:2000 |
| Peptide, recombinant protein | Streptavidin, Alexa-647 conjugated | Jackson | 016-600-084, RRID: AB_2341101 | 1:2000 |
| Commercial assay or kit | VECTASTAIN Elite ABC-HRP Kit, Peroxidase (Standard) | Vector Laboratories | PK-6100, RRID: AB_2336819 | 1:300 |
| Chemical compound, drug | 0.5% Cholera Toxin B subunit | List Biological Laboratories | Cat.: 104 | |
| Chemical compound, drug | 2% Fluoro-Gold | Fluorochrome LLC | | |
| Chemical compound, drug | Rimadyl (Carprofen 1.4 mg/kg) | Pfizer | | |
| Chemical compound, drug | 4% paraformaldehyde | Sigma-Aldrich | CAS No. 30525-89-4 | |
| Chemical compound, drug | 5% biotinylated dextrane amine | Molecular Probes | D1956, RRID: AB_2307337 | Molecular weight: 10,000 |
| Chemical compound, drug | Triton-X | Sigma-Aldrich | CAS Number: 9036-19-5 | IF: 0.5% DAB-Ni: 0.2% |
| Chemical compound, drug | 3-3'-diaminobenzidine | Sigma-Aldrich | CAS Number: 91-95-2 | |
| Software, algorithm | CaseViewer 2.4 | 3DHistech, Hungary | | |
| Software, algorithm | Zeiss ZEN 2010B SP1 Release version 6.0 | Zeiss Microimaging GmbH | | |
| Software, algorithm | CellSens Entry 1.16 | Olympus Corporation | | |
| Software, algorithm | ImageJ | NIH | | |
| Software, algorithm | Axon density analyzer script | *Mátyás et al., 2018* | | Available at: https://github.com/baabek/Axon-density-analyzer-ImageJ-script.git. |
| Software, algorithm | SPSS Statistics, ver. 27.0.1.0 | IBM | | |
| Other | Vectashield | Vector Laboratories | H-1000, RRID: AB_2336789 | Antifade mounting medium for fluorescent samples (see Materials and methods/Fluorescent immunohistochemistry) |
| Other | DePex | Serva, Germany | Cat. No. 18243 | Mounting medium for histological samples (see Materials and methods/Immunoperoxidase staining) |

