## [Editor Report]

This study provides valuable and detailed information regarding the connectivity between the medial prefrontal cortex (mPFC) and two major projection targets, the nucleus accumbens (NAc) and the ventral tegmental area (VTA). The authors show that mPFC neurons projecting to the NAc and VTA form distinct, largely non-overlapping cell groups characterized by distribution patterns in mPFC, their layers, and gene expressions. The authors also identify useful molecular markers for these populations. Overall, this study provides a valuable and solid resource with which to investigate neural circuits involved in motivated behaviors.

---

## [Decision Letter]

**Decision letter after peer review:**

Thank you for submitting your article "Molecular characteristics and laminar distribution of prefrontal neurons projecting to the mesolimbic system" for consideration by *eLife*. Your article has been reviewed by 2 peer reviewers, and the evaluation has been overseen by a Reviewing Editor and Kate Wassum as the Senior Editor. The reviewers have opted to remain anonymous.

Essential revisions:

This study examined the nature of projections from the prefrontal cortex (PFC) to the nucleus accumbens (NAc) and to the ventral tegmental area (VTA). The authors show that PFC projections to NAc and VTA are largely non-overlapping, originate in different layers of PFC, and express different molecular markers. The reviewers thought that this study provides high-quality data to the long-standing question. The reviewers thought, however, that the novelty of the study or controversies in the field need to be discussed more carefully, and the relation to recent published works should be discussed in a fair manner. Please address each concern noted in the public reviewers and recommendations for authors and especially the following essential revisions.

1. We would like to ask for more careful framing. There are many places throughout the manuscript where the authors claim that there is a great deal of controversy about the extent of the branching of these neurons. That is true, if you ask about *all* mPFC neurons. However, the vast majority of their citations either do not look at mPFC->NAc or do not look at mPFC->VTA. Instead, they look at mPFC projections to other cortical regions, thalamus, amgydala, etc. The contradiction, therefore, does not appear to be as deep as what they suggest. It would be best to reframe those portions of the paper just about prior evidence for mPFC->NAc/mPFC->VTA neurons, not all of the others. Please see more in Reviewer 1's comment.

2. Reviewer 2 raises some issues in potential overlaps with a few recent large-scale studies. We do not necessarily think that these studies reduce the value of the present work, but it would be helpful to discuss the relationship between the present and these studies.

---

## [Author Response]

Essential revisions:This study examined the nature of projections from the prefrontal cortex (PFC) to the nucleus accumbens (NAc) and to the ventral tegmental area (VTA). The authors show that PFC projections to NAc and VTA are largely non-overlapping, originate in different layers of PFC, and express different molecular markers. The reviewers thought that this study provides high-quality data to the long-standing question. The reviewers thought, however, that the novelty of the study or controversies in the field need to be discussed more carefully, and the relation to recent published works should be discussed in a fair manner. Please address each concern noted in the public reviewers and recommendations for authors and especially the following essential revisions.1. We would like to ask for more careful framing. There are many places throughout the manuscript where the authors claim that there is a great deal of controversy about the extent of the branching of these neurons. That is true, if you ask about *all* mPFC neurons. However, the vast majority of their citations either do not look at mPFC->NAc or do not look at mPFC->VTA. Instead, they look at mPFC projections to other cortical regions, thalamus, amgydala, etc. The contradiction, therefore, does not appear to be as deep as what they suggest. It would be best to reframe those portions of the paper just about prior evidence for mPFC->NAc/mPFC->VTA neurons, not all of the others.

We agree with our reviewer that some of our statements were misleading. Thus, we have rephrased the Introduction (Pages 3-5; Lines 54-99), Results (Page 20; Lines 356-363) and Discussion (Pages 26-32; lines 472-478, 492-498, 525, 530-532, 551-578, 586, 596-602) sections to focus on the controversial issues of the simultaneous projection to NAc and VTA by the same prefrontal cortical population.

2. Reviewer 2 raises some issues in potential overlaps with a few recent large-scale studies. We do not necessarily think that these studies reduce the value of the present work, but it would be helpful to discuss the relationship between the present and these studies.

The Gao et al. (2022) study published during our reviewing process, indeed, confirms some of our findings. A common finding of ours and Gao et al. (2022) is that mPFC_NAc_ and mPFC_VTA_ neurons form distinct classes within the mPFC projecting neuronal population. In addition, there is a small PT-like mPFC population, located rather in the L5b (showing Rbp4 expression in our study), which sends branching axons to both NAc and VTA.

Still, we think that the novelty of our work has remained significant.

i) We have provided easy-to-use, widely available molecular approaches to investigate mPFC territories and laminar organization. Using the detailed expression pattern of neurochemical markers revealed via multiple immunohistochemical technique and confocal microscopy, we were able to delineate borders between mPFC regions and layers with considerably high precision. For this purpose, mostly brain atlases are used. However, in a cortical region, like the mPFC, territory borders and shapes, as well as laminar thickness and depth are greatly changing at antero-posterior as well as dorso-ventral axes. Therefore, experiment-to-experiment, ‘stable’ markers are necessary to identify the exact location of neurons, recording sites, optic fibre positions, etc; that we, in our opinion, provided in the present study.

ii) Using the presented direct molecular composition, we have identified genetic markers for selective examination of layer- (and at some extent, region-)specific mPFC_NAc_ and mPFC_VTA_ populations.

iii) We have also provided evidence for the utility of this characterization using Cre mouse lines. The use of *Calb1-*, *Rbp4-*, *Ntsr1-* and *FoxP2-Cre*, which strains are widely used in cortical studies, in an intersectional approach, allow scientist to selectively investigate each of these mPFC populations, even in a target-selective manner via an intersectional approach.